# Differentiation of Therapeutic and Illicit Drug Use via Metabolite Profiling

**DOI:** 10.3390/metabo15110745

**Published:** 2025-11-17

**Authors:** Stanila Stoeva-Grigorova, Nadezhda Hvarchanova, Silvia Gancheva, Miroslav Eftimov, Kaloyan D. Georgiev, Maya Radeva-Ilieva

**Affiliations:** 1Department of Pharmacology, Toxicology and Pharmacotherapy, Faculty of Pharmacy, Medical University of Varna, 9000 Varna, Bulgaria; stoeva.st@mu-varna.bg (S.S.-G.); nadejda.hvarchanova@mu-varna.bg (N.H.); kaloyan.georgiev@mu-varna.bg (K.D.G.); 2Department of Pharmacology and Clinical Pharmacology and Therapeutics, Faculty of Medicine, Medical University of Varna, 9002 Varna, Bulgaria; silvia.gancheva@mu-varna.bg (S.G.); miroslav.eftimov@mu-varna.bg (M.E.)

**Keywords:** illicit drugs, controlled substances, opioids, stimulants, benzodiazepines, misuse, metabolites, metabolite profiling, toxicology

## Abstract

**Objectives**: The therapeutic use of controlled substances, particularly opioids, stimulants, and benzodiazepines, has significantly increased in recent decades. This is often accompanied by non-medical use and diversion, posing challenges for healthcare professionals and forensic experts monitoring potential misuse. As a result, the blurred boundary between legitimate therapy and substance abuse complicates the interpretation of toxicological results in clinical, legal, and occupational contexts. **Methods**: This review summarizes recent strategies for distinguishing therapeutic from illicit drug use through the analysis of substances and their metabolites in biological samples using sensitive and specific analytical methods. **Results**: Traditional drug abuse testing methods, based on parent substance detection, often lack the specificity needed to differentiate therapeutic use from illicit intake. Therefore, advanced analytical methods are required to accurately differentiate the source, route, and adherence to therapy. Therapeutic and illicit forms of the same substance can exhibit distinct metabolic profiles, with certain metabolites serving as biomarkers for illicit drug use. In some cases, chiral analysis may also aid in determining the drug source. Other studies have shown that the ratio of the parent compound to its metabolites (or between different metabolites) may reflect the pattern of use, such as chronic versus acute use or the route of administration. Illicit drugs may also contain synthesis by-products or cutting agents, detectable through advanced techniques. **Conclusions**: Metabolite profiling offers a robust approach for differentiating therapeutic from illicit drug use and is expected to be increasingly applied in clinical toxicology, forensic investigations, workplace testing, and/or doping control.

## 1. Introduction

The global increase in the therapeutic use of controlled substances, particularly opioids, benzodiazepines, stimulants, cannabinoid-based medicines has paralleled a troubling rise in their non-medical use, diversion, and associated morbidity [1,2]. While these drugs remain essential for managing pain, anxiety, sleep disorders, and other conditions, their misuse has become a major public health and forensic concern. The overlapping biochemical signatures of legitimate therapeutic intake and illicit consumption often blur the boundary between compliance and abuse, creating interpretative challenges in clinical toxicology, workplace testing, and legal settings [3].

Traditional toxicological screening methods, largely based on immunoassays, detect parent compounds or specific metabolites but often lack the specificity required to discriminate between therapeutic dosing, chronic misuse, and illicit analog consumption [4]. The development and application of chromatographic and mass spectrometric techniques such as gas chromatography-mass spectrometry (GC–MS), liquid chromatography-tandem mass spectrometry (LC–MS/MS) and high-resolution mass spectrometry (HRMS) has transformed drug analysis by enabling the precise identification of drug-specific metabolic pathways and biomarkers [5]. These advancements now allow differentiation between pharmaceutical formulations and clandestine preparations, offering insight into both the source and metabolic fate of a given substance [6]. Among opioids, the distinction between therapeutic administration (e.g., morphine, oxycodone) and illicit use (e.g., heroin, fentanyl analogs) can be elucidated through unique metabolite markers such as 6-monoacetylmorphine or norfentanyl [7]. For benzodiazepines and Z-drugs, complex metabolic interconversion such as between diazepam, nordiazepam, and oxazepam, necessitates careful interpretation of parent/metabolite ratios and matrix-specific detection windows [8,9]. Similarly, stimulant analysis increasingly relies on enantiomeric profiling to differentiate legitimate amphetamine-based medications from illicit methamphetamine or designer analogs [10]. In the case of cannabinoids, advanced metabolomic and isotopic approaches can distinguish pharmaceutical formulations (e.g., dronabinol, nabiximols) from recreational cannabis exposure, even in chronic users [11].

Obviously, the metabolite profiling has emerged as a cornerstone strategy for differentiating therapeutic from illicit drug use. By integrating quantitative metabolite ratios, metabolic fingerprinting, and multi-matrix testing (e.g., plasma, urine, hair, oral fluid), it is possible to achieve a nuanced understanding of drug exposure patterns, metabolic dynamics, and compliance with prescribed therapy [12]. This article provides a comprehensive overview of analytical and interpretative approaches for clarifying the often-ambiguous boundary between medical treatment and abuse with key controlled substance classes. The main objective of this review is to provide a critical synthesis of contemporary evidence regarding the utility of metabolite profiling in differentiating therapeutic application from illicit consumption.

## 2. Materials and Methods

In order to gain a better understanding of differentiating therapeutic from illicit drug use through metabolite profiling, we conducted an in-depth literature review on the Web of Science, Scopus, PubMed and ResearchGate databases. The initial search yielded over 2000 literature sources related to the topic. In result, we selected over 300 articles, mostly published in the last fifteen years, and comprehensively analyzed the available scientific information. Both review and research articles written in English were cited in the manuscript. Additionally, scientific information was retrieved from the internet databases of the European Medicine Agency (EMA), European Union Drugs Agency (EUDA) and other organisations.

## 3. Opioids

Opioid agonists include natural opiates, semisynthetic alkaloids derived from the opium poppy, and fully synthetic compounds that act via the same mechanism [13]. While some of these substances were originally developed for medical purposes, certain agents, such as heroin, eventually became primarily used for recreational purposes. Opioid receptors, which are G protein–coupled, mediate the effects of both endogenous ligands (e.g., endorphins) and exogenously administered opioids. The three main types of classical opioid receptors are μ (mu), δ (delta), and κ (kappa) [13,14]. Interaction with these receptors can elicit agonist, partial agonist, or antagonist responses. In clinical settings, most opioids exert an agonistic effect, producing analgesia [15]. They remain a primary therapeutic option for managing short- to intermediate-duration moderate-to-severe pain, including in oncological practice. Certain opioids are also employed as anesthetic adjuvants and as antitussive agents [16]. Prolonged opioid use, however, is limited by the risk of multiple adverse effects, the most serious being respiratory depression. Other notable manifestations include constipation, nausea, pruritus, and paradoxical hyperalgesia (enhanced pain sensitivity). Additionally, opioids induce molecular adaptive changes that reduce their efficacy and contribute to the development of dependence, highlighting the complex and nuanced safety profile of these agents [17]. Moreover, due to their pronounced euphoric effects, μ-opioid agonists are among the most frequently abused substances, with substantial health, social, and economic consequences [13].

In response to the profound psychosocial consequences associated with opioid (mis)use, the fifth edition of the Diagnostic and Statistical Manual of Mental Disorders (DSM-5, American Psychiatric Association) introduced the term Opioid Use Disorder (OUD). This designation characterizes the chronic use of opioids that results in clinically significant distress or functional impairment [18]. The clinical presentation is marked by an irresistible craving for opioids, progressive tolerance, and the emergence of withdrawal symptoms upon cessation of use. Consequently, the spectrum of OUD encompasses conditions ranging from subclinical dependence to full-blown opioid addiction [19,20]. In this context, the opioid crisis remains an urgent global challenge, with the current phase frequently described as the “fourth wave,” characterized by epidemic proliferation, including combined use with stimulants. The prevalence of comorbid psychiatric disorders is markedly higher compared to previous phases [21,22]. Data from the World Health Organization (WHO) indicate that of roughly 600,000 drug-related deaths recorded globally, approximately 80% involve opioids, with nearly one-quarter resulting directly from overdose [23]. Forecasts suggest that by 2029, as many as 1.2 million people in the United States and Canada alone could succumb to opioid overdoses [24]. These statistics reflect not only heroin misuse but also the extensive and often inappropriate consumption of prescription opioid analgesics [25]. Given that this public health crisis has recently affected other regions, including Australia and Europe, official figures likely significantly underestimate the true magnitude of the epidemic [25,26,27].

The opioid medications most commonly misused include hydrocodone, oxycodone, methadone, buprenorphine, fentanyl, tramadol, and morphine [28,29,30]. Even in cases of nonmedical use, the preferred route of opioid administration remains oral, followed by intravenous and intranasal routes, with buccal administration being considerably less common [25,31,32,33]. Currently, therapeutic interventions for managing opioid dependence include buprenorphine, methadone, and extended-release naltrexone, as well as the α_2_-adrenergic agonist lofexidine to mitigate the acute manifestations of opioid withdrawal [20,34]. Despite the availability of treatment options, the most recent United Nations Office on Drugs and Crime report indicates that fewer than 20% of affected individuals globally have access to adequate care. In recent years, it has become increasingly evident that the primary drivers of OUD are substances legally approved for medical use. Estimates suggest that approximately 60 million individuals engaged in nonmedical opioid use in 2021. These data underscore the urgent need to reassess strategies for biomonitoring opioid use while ensuring that the control of illicitly manufactured or diverted pharmaceutical opioids is not neglected [35].

In contemporary practice, toxicochemical analysis of opioids has emerged as an essential tool, extending far beyond its traditional role in toxicology and forensic investigations. Its utility now spans occupational health, military oversight, and sports medicine, where results can directly shape both clinical management and legal outcomes. Within pain management programs, systematic opioid testing, most often via urine analysis has become pivotal for verifying adherence to prescribed therapy and for preventing diversion or substitution of medications. These realities underscore the critical need to distinguish medical use from non-medical opioid consumption with precision and reliability [29,36].

Opioid detection mainly employs immunoassays, which are inexpensive but prone to cross-reactivity [37]. Chromatography coupled with MS remains the gold standard for confirmation, offering superior specificity despite higher complexity and cost [38,39]. In recent years, Mayo Clinic laboratories have advanced opioid analysis with the implementation of high-resolution targeted opioid screening (TOSU), leveraging LC-MS/MS. This approach enables the qualitative identification of 33 distinct opioids and their metabolites in urine without the need for hydrolysis, providing clinicians with a powerful tool to monitor therapeutic compliance and detect unauthorized use (Table 1). Such innovations not only enhance diagnostic accuracy but also reflect the evolving landscape of opioid monitoring, where precision is paramount for both patient care and public health [37,40]. Recent studies combine untargeted ultra high performance liquid chromatography (UHPLC)-HRMS metabolomics with machine learning classifiers to detect characteristic exposure signatures and to differentiate natural opioids (for example morphine or codeine) from potent synthetic analogues such as fentanyl derivatives. These workflows often rely on endogenous metabolomic perturbations as indirect biomarkers when parent new synthetic opioid compounds are present at very low concentrations in biological matrices. Such combined mass spectrometry–machine learning strategies substantially improve screening sensitivity for novel opioids and help prioritize candidate metabolites for subsequent tandem mass spectrometry (MS/MS) confirmation [41].

In addition to a thorough command of laboratory analytical techniques, a deep understanding of opioid metabolism is indispensable when interpreting biological specimens—most commonly urine and blood. Many clinically used opioids undergo metabolic conversion into other active opioids and/or may appear as “manufacturing-related impurities”. If such transformations are overlooked, they may yield misleading conclusions regarding patient misuse or drug exposure [39]. Opioid metabolism occurs mainly in the liver through Phase I (cytochrome P450 (CYP)-mediated oxidation/reduction) and Phase II (glucuronidation) reactions. Variability in metabolism arises from CYP2D6 and CYP3A4 activity, influenced by drug interactions and genetic polymorphisms [42]. A key example is the CYP2D6-mediated O-demethylation of codeine to morphine, which accounts for about 10% of the administered dose and underlies codeine’s analgesic effect (Figure 1). Ultra-rapid metabolizers may generate toxic morphine levels, while similar pathways convert oxycodone and hydrocodone to oxymorphone and hydromorphone, respectively. At higher doses, secondary metabolites such as hydrocodone and hydromorphone may appear, complicating toxicological analysis. In contrast, opioids like buprenorphine, fentanyl, meperidine, methadone, and tramadol are metabolically independent, exhibiting distinct pharmacokinetic and toxicological profiles that demand precise analytical characterization [39,43].

### 3.1. Morphine/Codeine/Diacethylmorphine (Heroin)

In most conventional immunoassays designed to detect opioid misuse, the prototypical opioid, morphine, is employed as the sole calibrator. Beyond serving as a direct analyte for its own detection, morphine also functions as an indirect biomarker of codeine, heroin, and ethylmorphine misuse, owing to its role as their principal metabolite [29]. In clinical practice, a threshold of 300 ng/mL (morphine equivalents) is commonly applied for opioid screening, a level sensitive to morphine and codeine but potentially insufficient for detecting other opioids. For federal and workplace drug testing, however, substantially higher cut-off values have been established—2000 ng/mL for screening and 4000 ng/mL (morphine)/2000 ng/mL (codeine) for confirmatory testing. These thresholds were introduced to minimize the likelihood of false-positive results, such as those arising from the consumption of poppy seeds or the legitimate therapeutic use of prescribed opioids during heroin abuse screening. Consequently, such stringent cut-offs may not always be appropriate for clinical diagnosis or therapeutic monitoring [42].

A major limitation of employing morphine as a biomarker of opioid misuse lies in the fact that this approach cannot distinguish between pharmaceutical-grade heroin and illicit street heroin [44]. Moreover, opioid misuse is not confined to morphine-like compounds, which increases the risk of false-negative results for certain semisynthetic and synthetic agents. At the same time, several commercially available immunoassays exhibit cross-reactivity with semisynthetic opioids. In addition, morphine-3-glucuronide, codeine, and 6-acetylmorphine may also react with morphine-specific antibodies, thereby contributing to the reporting of positive screening results [45]. Advanced immunoassays and LC-MS/MS methods provide substantially greater specificity, yet remain unavailable in many toxicology laboratories. An important step toward improving chromatographic separation in confirmatory testing of natural opioids is the preliminary hydrolysis of glucuronide conjugates, followed by derivatization of the parent compounds [29,46].

Codeine itself is not formed as a metabolite following the use of morphine, ethylmorphine, or pharmaceutical heroin. However, in addition to its presence after the administration of codeine-containing medications, it may also appear in the body after the consumption of street heroin—where it represents a metabolite of one of its common impurities—as well as after ingestion of poppy seed products or opium preparations [47]. For the detection of codeine use, in addition to monitoring its principal metabolite, morphine, the morphine/codeine (M/C) ratio can also be applied. Generally, an M/C ratio of <1 is considered indicative of codeine use alone, whereas a ratio of >1 suggests exposure to morphine or heroin. According to Ellis et al. (2016), an M/C ratio exceeding 1 in intravenous drug users constitutes a reliable indicator of fatal heroin intoxication, even in the absence of 6-MAM, particularly when combined with scene and autopsy findings confirming the heroin origin of morphine and codeine [48]. Nevertheless, this numerical threshold should not be regarded as absolute. In individuals with ultra-rapid CYP2D6 metabolism, the M/C ratio may exceed 1 even following codeine intake alone, underscoring the complexity of metabolic interactions and the necessity for cautious interpretation of toxicological results [49]. To support the reliable differentiation between opium/codeine users and heroin users, Al-Amri et al. proposed neopine as a biomarker. Neopine is detectable in opium, and consequently in the urine of opium and codeine users as well as after poppy seed ingestion, but is absent in heroin and in heroin users. Nonetheless, the possibility of combined use or interference from poppy-derived products cannot be fully excluded [50].

The short half-life of heroin, combined with its poor stability in aqueous environments, renders its direct detection in biological specimens both laborious and highly unsuitable as a reliable confirmation of use [51]. Once introduced into the body, heroin undergoes rapid enzymatic hydrolysis by esterases, yielding 6-MAM within minutes. This metabolite is a unique product of illicit heroin metabolism and is widely regarded as the gold standard biomarker for its monitoring (cut-off: 10 ng/mL). As a precursor to morphine, 6-MAM holds exceptional forensic value; however, its short elimination half-life narrows its window of detection to approximately 6–8 h, with optimal laboratory conditions extending this period to no more than 24 h post-consumption. Consequently, 6-MAM serves as a highly specific marker of very recent heroin use [52,53,54]. Nonetheless, its utility is limited. Detection of 6-MAM is challenging following oral or rectal administration of heroin due to extensive first-pass metabolism to morphine, and it cannot distinguish between pharmaceutical-grade diacetylmorphine—still in use in certain countries—and illicit street heroin [47]. In such scenarios, reliance solely on 6-MAM is insufficient for forensic differentiation. Instead, the analysis of additional opioid derivatives or their metabolites in blood or urine may provide stronger correlations with illicit formulations [51]. One such biomarker is meconin, a metabolite of noscapine—the second most abundant alkaloid in opium after morphine, and frequently encountered as an impurity in street heroin [55,56,57]. Accordingly, meconin should not be present in pharmaceutical-grade heroin [58]. Supporting this, Jones et al. (2015) developed an LC-MS/MS method for the simultaneous quantification of codeine, morphine, 6-MAM, and meconin in umbilical cord tissue, demonstrating that meconin can serve as a biomarker of in utero heroin exposure [59]. Caution, however, is warranted when interpreting meconin detection, as noscapine is also found in antitussive formulations, and meconin itself has been identified as a component of the herbal supplement “Golden Seal”, commonly marketed as a dietary additive [60,61,62].

Another secondary biomarker arises from chemical impurities generated during illicit heroin synthesis, namely 6-acetylcodeine (6-AC). This compound is a synthetic by-product formed during the acetylation of morphine in opium extracts [46,63]. It has no therapeutic application and is rarely observed following the use of prescribed opioids. Some sporadic reports indicate that 6-AC can also be formed after the simultaneous administration of high intravenous doses of heroin (120–400 mg) and oral codeine (e.g., 50 mg as an antitussive) [64]. Accordingly, the presence of 6-AC in biological specimens, particularly when detected alongside other characteristic impurities, is generally interpreted as a specific indicator of illicit heroin use rather than legitimate therapeutic opioid intake [65]. Similar to 6-MAM, 6-AC can be detected in urine for approximately 8 h, with codeine as its principal metabolite. However, numerous studies have shown that 6-AC is a less robust and less reliable marker of heroin use compared to 6-MAM, due to its lower peak concentrations and short half-life (~237 min, yielding a diagnostic “window” of ~8 h). As a result, there is a high likelihood that biological samples may test negative for 6-AC despite the presence of 6-MAM. For this reason, 6-AC does not yet have practical value as a primary biomarker, serving instead as supplementary evidence of heroin exposure [66].

Consumption of poppy seed-containing foods can elevate urinary opioid levels above 300 ng/mL—the standard positive threshold for immunoassay-based screening still employed in some laboratories [42,63]. In forensic practice, this phenomenon has been invoked in the so-called “poppy seed defense”, which has led to acquittals in cases of alleged opioid misuse and subsequently motivated the adoption of a higher positive screening cut-off of 2000 ng/mL. In a recent study, Reisfield et al. (2023) highlighted that accurate differentiation between individuals using opioids and those exposed through dietary intake requires consideration of several factors: the specific food product, portion size, pattern of consumption, the time elapsed between ingestion and urine collection, and, when possible, analysis of the particular seed batch consumed [67]. In this context, an official 2024 publication by the U.S. Department of Defense noted that the military’s drug testing program has raised the codeine cut-off to reduce false-positive results among service members. Additionally, the program has introduced a reflex screening test for the opiate thebaine, a minor alkaloid derived from the poppy plant, which can serve as a urinary biomarker of poppy seed consumption. In the pharmaceutical industry, thebaine is employed as a precursor in the synthesis of oxycodone, oxymorphone, naloxone, and buprenorphine [68,69]. It is not available as a therapeutic product and is found only in trace amounts in street heroin—if at all—because it is typically converted into acetylated derivatives alongside morphine [47]. Consequently, some authorities classify urine samples containing 4000–10,000 ng/mL of codeine as negative for opioid misuse, provided that thebaine is also detected at concentrations of ≥ 5 ng/mL, reflecting dietary exposure rather than illicit use [70].

In recent years, attention has focused on the acetylated metabolite of thebaine-4-glucuronide, which is formed during the illicit synthesis of heroin, when thebaine from opium undergoes acetylation and chemical restructuring. Paradoxically, the presence of this thebaine metabolite serves as a forensic indicator of street heroin use, as it is not observed following consumption of poppy seeds or pharmaceutical opioids. It is excreted in the urine and results from the glucuronidation of acetylated thebaine derivatives generated during illicit heroin production. This metabolite is increasingly recognized for its diagnostic value as a forensic biomarker in modern toxicology laboratories [71]. Another potential marker is oripavine, produced via O-demethylation of thebaine [72]. Since thebaine is destroyed during acetylation in heroin synthesis, neither thebaine nor oripavine is expected after consumption of street heroin or pharmaceutical phenanthrene-based morphinomimetics [55]. Therefore, the detection of thebaine and/or oripavine alongside morphine in urine indicates opium or poppy seed intake, though it does not exclude concurrent use of other opioids [73].

According to Maas et al. (2018) noscapine and papaverine are not detectable in urine following the consumption of poppy-derived products containing up to 94 µg of noscapine and 3.3 µg of papaverine [47]. However, desmethylpapaverine and its glucuronide can be detected for up to 48 h post-ingestion [74]. Consequently, these compounds may serve as potential biomarkers of poppy consumption, provided that the possibility of their origin from pharmaceutical products is excluded [63]. Reticuline, a precursor of opium alkaloids, is absent in both heroin and poppy seeds, being present exclusively in opium. Therefore, it has been proposed as a biomarker to differentiate opium use from dietary poppy exposure, as well as to distinguish opium intake from pharmaceutical codeine use [63,75].

Considering the summarized advances in the detection of primary and secondary biomarkers of heroin use, particularly intriguing are efforts to implement comprehensive multi-biomarker analyses in biological specimens. A notable example is the GC-MS method developed by Karakasi et al. (2024), which allows simultaneous determination of five potential heroin biomarkers (meconin, thebaine, papaverine, acetylcodeine, and noscapine), together with the classical opioids morphine, codeine, and 6-MAM, in a variety of biological matrices, including blood, urine, vitreous humor, and bile, from heroin users [51]. This approach aims to provide a more in-depth characterization of heroin exposure, particularly in cases where direct evidence of heroin use is sought. The authors concluded that meconin, primarily, and papaverine, secondarily, could serve as indirect biomarkers of heroin use in biological materials, especially in scenarios where 6-MAM has already been metabolized and is no longer detectable [51]. Using a similar analytical strategy, Al-Asmari et al. emphasized that in postmortem biomarking of heroin, the determination of free morphine in solid tissues is more reliable and informative. In particular, the gastric wall shows a higher frequency of 6-MAM detection compared to other tissues, highlighting the utility of alternative matrices in cases where blood samples are unavailable [76].

### 3.2. Oxycodone/Oxymorphone/Hydrocodone/Hydromorphone

Oxycodone is well absorbed following oral administration and exhibits a relatively rapid onset of action. Its duration of effect ranges from 4 to 12 h, depending on the formulation. Oxycodone undergoes metabolism via N- and O-demethylation, resulting in the formation of the active metabolite oxymorphone, which is also available as a therapeutic agent for pain management. The majority of the administered dose is excreted in urine as free oxycodone and noroxycodone (with moderate activity), as well as conjugated oxycodone and oxymorphone [28]. Significant interindividual variability in metabolism arises from genetic polymorphisms [46]. Both oxycodone and oxymorphone demonstrate low cross-reactivity with standard opioid immunoassays, and therefore are often undetectable through routine opioid screening tests [29]. Specific immunoassays have been developed for oxycodone due to its high potential for abuse [45]. Hydrocodone is structurally similar to oxycodone. At lower doses, it is used as an antitussive, while at higher doses it is prescribed as an analgesic. Its duration of action ranges from 4 to 6 h, and it is approximately six times more potent than codeine. Like oxycodone, hydrocodone undergoes N- and O-demethylation, producing hydromorphone, an active metabolite approximately 8.5 times more potent than morphine, as well as dihydrocodeine via reduction in the ketone group [28].

Although hydromorphone is a minor metabolite of morphine, oxymorphone is not a metabolite of either morphine or hydromorphone. Therefore, the presence of oxymorphone in the absence of a valid prescription is a clear indicator of opioid misuse. Hydrocodone can also arise as a minor metabolite of morphine and may constitute up to 11% of the codeine concentration. Consequently, the detection of small amounts of hydrocodone in urine with high codeine levels should not be interpreted as evidence of hydrocodone misuse [38]. Most routine opioid immunoassays exhibit weak cross-reactivity with oxycodone and relative cross-reactivity with hydromorphone and hydrocodone. At high urinary concentrations, however, these compounds may still yield positive results, justifying the development of specific immunoassays for oxycodone due to its widespread abuse [45]. Overall, keto-opioids undergo extensive conjugation, necessitating hydrolysis and derivatization for accurate quantitative analysis [28,45].

### 3.3. Methadone

Methadone is primarily employed for maintenance treatment of opioid dependence, but it can also be used for pain management. It is widely available as a racemic mixture, although the (R)-enantiomer exhibits a tenfold higher affinity for μ- and δ-opioid receptors. Methadone is particularly effective for managing withdrawal symptoms due to its high oral bioavailability, long elimination half-life (up to 55 h), low severity of withdrawal upon discontinuation, and the availability of specific antagonists for overdose management [28]. The opioid is metabolized mainly in the liver via cytochrome P450 enzymes, predominantly CYP3A4 and, to a lesser extent, CYP2D6, as well as in the intestines. In approximately 10% of Caucasians, who are slow metabolizers with low CYP2D6 activity, the elimination half-life may be prolonged. Methadone and its primary metabolite, EDDP, are excreted in urine. Concomitantly, 2-ethyl-5-methyl-3,3-diphenylpyrrolidine (EDMP) is also formed, although it is quantitatively less significant (Figure 2) [77]. Measurement of both the parent drug and EDDP can be used to assess prolonged exposure, aiding in the differentiation of chronic therapeutic use from single or incidental exposure [78,79,80]. In clinical practice, it is important to measure both compounds because methadone excretion varies considerably depending on dose, metabolism, and urinary pH, whereas urinary EDDP levels are pH-independent and therefore preferred for monitoring treatment adherence [29,77].

Commercially available urinary immunoassays for methadone screening utilize antibodies targeting either methadone or EDDP. Typically, standard opioid immunoassays do not detect methadone or its metabolites, and methadone-specific assays may not detect EDDP and vice versa. However, immunoassays designed for methadone often exhibit low cross-reactivity with EDDP, necessitating confirmatory testing [45]. Additionally, the opioid analgesic tapentadol has been reported, albeit rarely, to cross-react with certain methadone immunoassays, potentially producing false-positive results, further highlighting the need for confirmatory analysis [42,81].

Another approach to distinguish the nature of methadone use is chiral analysis. Enantiomeric ratios of methadone and its metabolite EDDP can differentiate therapeutic from illicit use. For patients receiving racemic methadone therapy, the ratio is approximately 1:1, whereas therapy with pure R-methadone (available in some countries) lacks the S-enantiomer. Atypical enantiomeric profiles, inconsistent with approved formulations in a given jurisdiction, suggest illicit origin. Additionally, comparison of methadone and EDDP levels can indicate whether the drug has been metabolized in vivo or deliberately added to adulterate a sample [82,83,84]. Cases have been documented where patients intentionally fail to comply with prescribed opioid therapy for dependence treatment or pain management, sometimes diverting their medications to the illicit market. Routine drug screening is often a requirement for participation in such treatment programs to monitor compliance. To circumvent testing, some individuals employ the so-called “pill shaving” technique, which involves crushing or scraping opioid tablets (e.g., with a razor) into biological specimens to artificially generate a positive result during toxicological analysis [85,86]. Such attempts can be detected by comparing the relative amounts of parent methadone and EDDP. In cases of compromised adherence, the EDDP-to-methadone ratio is markedly low, sometimes below 0.09, revealing manipulation of the sample [29,87,88].

### 3.4. Fentanyl

Fentanyl is a synthetic opioid with a molecular structure distinct from that of morphine. First approved by the U.S. Food and Drug Administration (FDA) in the late 1960s, it is currently employed in pain management and anesthesia. However, the illicit use of fentanyl and its analogs has significantly intensified the opioid crisis in recent years, contributing to a sharp increase in overdose-related fatalities. Despite international regulatory efforts, fentanyl’s extreme potency and low production cost have facilitated widespread dissemination, including via counterfeit formulations. Although naloxone remains the primary antidote, its limited efficacy highlights the urgent need for more potent and long-acting therapeutic interventions. Additional risk factors, such as polypharmacy, prescription misuse, environmental exposure, and potential weaponization, further exacerbate public health challenges [89,90]. In this regard, analytical methodologies capable of distinguishing illicit from medically prescribed fentanyl use have become increasingly critical. A further concern is the emergence of novel fentanyl analogs on the illicit market. Many of these analogs are significantly more potent than fentanyl itself and are present at much lower doses, complicating their detection in blood and urine. This underscores the urgent need for accessible and reliable testing of these substances in both biological specimens and seized drugs [28].

Immunoassays for fentanyl in urine are available; however, their use is largely restricted to laboratories with appropriate licensure. Immunoassays have also been developed for alternative matrices, including oral fluid, blood, and meconium, but these are not widely accessible. Point-of-care tests, such as lateral flow immunoassays and fentanyl test strips, are more broadly available, yet their lower specificity limits clinical utility [91,92,93]. All immunoassays are susceptible to false-positive and false-negative results and vary in their ability to detect fentanyl analogs (e.g., carfentanil, acetylfentanyl, furanylfentanyl, etc.) and other novel synthetic opioids (NSOs) (e.g., U-47700, AH-7921, etc.) [91,94,95]. Confirmatory analysis is typically performed using mass spectrometry, which enables accurate and reliable detection of fentanyl and its primary metabolite, norfentanyl, in urine and oral fluid (Figure 3). High-performance liquid chromatography (HPLC)–MS/MS is the most widely employed method, offering exceptional specificity and sensitivity for fentanyl, norfentanyl, and various fentanyl analogs and NSOs [42,96,97,98,99].

In conclusion, it is important to recognize that certain medications, including over-the-counter products, can induce false-positive results in opioid immunoassays. For example, naloxone and its glucuronide metabolite may cross-react with antibodies in some CEDIA immunoassays at high concentrations. False-positive results for methadone have also been documented following administration of quetiapine and diphenhydramine [100,101]. Additionally, antibiotics such as fluoroquinolones (e.g., levofloxacin, ofloxacin, pefloxacin) and rifampicin may exhibit cross-reactivity in some opioid immunoassays [102]. These observations underscore the continued necessity of confirmatory analysis via chromatography to definitively rule out false-positive results [46]. Furthermore, assessing acceptable ranges for temperature, pH, creatinine concentration, and specific gravity of urine is essential for identifying adulterated, substituted, or diluted specimens [39,103]. Ultimately, all urine drug testing results must be interpreted in the context of the analytical method, prescribed medications, specimen type, time since last dose, specimen collection, specimen validity tests, and patient-specific factors. Unexpected or unexplained results should prompt discussion between the patient and laboratory, and additional confirmatory testing should be performed as needed to ensure accurate interpretation [37]. Table 2 summarizes key pharmacokinetic characteristics, analytical biomarkers, detection matrices, regulatory cut-off limits, and major analytical challenges associated with selected opioids [13,28,39,42,103,104,105].

## 4. Amphetamine-Type Stimulants

Amphetamine is the prototype of a broader class of compounds sharing a phenethylamine scaffold, collectively referred to as amphetamine-type stimulants (ATS). Small structural modifications, including substitution on the phenyl ring, α-methylation, β-ketone incorporation, or alteration of the amino group, generate a wide spectrum of ATS derivatives with diverse pharmacological profiles, ranging from classical stimulants to entactogens, hallucinogens, and monoamine oxidase inhibitors [106]. The group includes amphetamine, methamphetamine, and 3,4-methylenedioxymethamphetamine (MDMA), as well as synthetic cathinones such as mephedrone, methylone, and methcathinone [107,108].

Although these substances share a common structural backbone, they differ markedly in potency, pharmacokinetics, and clinical as well as toxicological effects. Several ATS were initially developed as pharmaceutical treatments for conditions such as narcolepsy, attention deficit hyperactivity disorder (ADHD), or as appetite suppressants. Today, legitimate medical use is mainly restricted to ADHD and narcolepsy. Amphetamine and mixed amphetamine salts are marketed as Adderall^®^, while methamphetamine hydrochloride is available as Desoxyn^®^ in select cases of severe ADHD or obesity. Lisdexamfetamine (Vyvanse^®^) is a prodrug that exclusively releases D-amphetamine, thereby reducing the risk of misuse compared to immediate-release formulations [108,109]. These pharmaceutical formulations are effective when used as prescribed, but they also pose a risk of diversion and abuse. Their strong psychostimulant and euphoric properties, in particular, have contributed to widespread nonmedical use [107,109].

Pharmacologically, ATS primarily target monoaminergic pathways. Acting as substrates for dopamine (DAT), norepinephrine (NET), and serotonin (SERT) transporters, and disrupting vesicular monoamine transporter 2 (VMAT2), they promote monoamine release and inhibit reuptake, thereby elevating extracellular neurotransmitter levels [110,111]. Therapeutically, such actions are beneficial for improving alertness and attention, but chronic or supratherapeutic exposure is linked to acute toxicity, neurotoxicity, tolerance, dependence, and psychosis [112,113]. Reflecting growing psychosocial impact, the fifth edition of the Diagnostic and Statistical Manual of Mental Disorders introduced Stimulant Use Disorder (StUD) [18]. This condition is characterized by compulsive consumption of amphetamines and related compounds, resulting in clinically significant distress or impairment. Clinical hallmarks include craving, tolerance, and increased risk of psychiatric and somatic complications, such as anxiety, psychosis, cardiovascular disease, and cognitive impairment [114]. Unlike opioid withdrawal, cessation of ATS is generally non-lethal, yet commonly associated with dysphoria, fatigue, sleep disturbances, and elevated relapse risk [115].

Epidemiologically, ATS are the second most commonly used illicit drug class after cannabis, with highest prevalence in East and Southeast Asia, Australia, North America, and parts of Europe. Methamphetamine-related harms, including overdose fatalities, psychotic episodes, and violent behavior, are sharply increasing worldwide. Notably, beyond psychiatric and somatic harms, ATS, particularly methamphetamine and MDMA, have also been implicated in drug-facilitated sexual assault (DFSA) due to their ability to induce disinhibition, hypersexuality, and impaired judgment. This forensic dimension underscores their dual role as drugs of abuse and potential tools in sexual crimes [116,117]. Polysubstance use with opioids contributes to the ongoing “fourth wave” of the opioid crisis [107,118].

The ATS most frequently misused include amphetamine, methamphetamine, 3,4-methylenedioxymethamphetamine (MDMA), and synthetic cathinones such as mephedrone and methylone [107]. Patterns of nonmedical use vary, with the preferred routes of administration being oral and intranasal, followed by smoking or intravenous injection in the case of methamphetamine, which is associated with particularly severe health risks [114].

Currently, therapeutic interventions specifically approved for ATS use disorder remain limited. Management is based primarily on psychosocial interventions, including cognitive-behavioral therapy and contingency management, with no pharmacological agent yet demonstrating consistent efficacy [118]. Despite intensive research, candidate medications such as bupropion, mirtazapine, and modafinil have shown only mixed or modest results [119].

In parallel, the modes of ATS acquisition represent a major challenge, encompassing large-scale illicit production and trafficking, diversion of pharmaceutical preparations, and increasingly, internet-based markets. These supply channels contribute substantially to the persistence and expansion of ATS use worldwide [107,120,121]. Diversion of prescription stimulants for nonmedical use is also a growing concern, particularly in high-income countries [122]. Documented cases describe the extraction and misuse of active compounds from pharmaceutical preparations, such as crushing and intranasal administration of Adderall^®^ tablets or chemical isolation of methamphetamine from Desoxyn^®^. These practices not only enhance rapid onset of psychoactive effects but also increase the risk of overdose and rapid dependence development [123]. In summary, ATS represent a dual-edged phenomenon: indispensable in selected medical contexts yet carrying immense risks of diversion, dependence, and severe societal harms.

Collectively, these findings underscore a dual challenge: the continued global availability of ATS through illicit and diverted channels, and the lack of effective pharmacological therapies. Addressing this issue will require comprehensive strategies that combine improved biomonitoring of stimulant use with strengthened control of illicit manufacturing and pharmaceutical diversion [118,119].

In modern clinical and regulatory practice, the toxicochemical analysis of amphetamines has evolved far beyond its original role in forensic and toxicology laboratories. Today, ATS testing is applied in diverse settings including occupational health, military medicine, and sports, where analytical results directly influence both medical management and regulatory compliance [124,125]. Within psychiatry and addiction medicine, routine urine and hair testing has become central to monitoring therapeutic adherence, detecting relapse, and identifying unprescribed stimulant use [46,126]. At the population level, incorporation of ATS biomarkers into wastewater-based epidemiology has enabled near real-time surveillance of consumption trends, providing valuable insight into community-level stimulant use. In particular, the detection of non-physiological by-products such as paramethoxyamphetamine (PMA) or paramethoxymethamphetamine (PMMA), together with chiral analysis differentiating enantiomerically pure pharmaceutical formulations from racemic illicit preparations, represents a forensic “gold standard” for discriminating legitimate therapeutic exposure from illicit ATS use [127,128,129]. Taken together, these advances highlight the urgent need for validated methodologies capable of distinguishing medically indicated stimulant therapy from illicit or nonmedical consumption. Overall, advances in analytical mothodologies ensure more reliable differentiation between therapeutic and illicit use of ATS, thereby strengthening clinical decision-making and enhancing forensic validity. Beyond urine, other biological matrices offer complementary analytical value as oral fluid enables near real-time detection and is minimally affected by urinary pH. Hair testing provides retrospective monitoring over extended periods and is particularly useful in compliance assessment and forensic investigations whereas plasma and serum are primarily applied in cases of acute intoxication or for medicolegal reconstruction [124,125,126]. An overview of the recommended interpretive workflow for ATS testing, from initial screening to forensic attribution, is illustrated in Figure 4.

Selected parent/metabolite ratios (e.g., amphetamine/methamphetamine) help infer recency of use and source attribution when interpreted alongside urine pH, dose, and clinical context [113,130]. For 3,4-methylenedioxymethamphetamine, the ratio of the parent compound to its metabolite 3,4-methylenedioxyamphetamine is used as a temporal marker: a high MDMA/MDA ratio suggests recent ingestion, whereas a decreasing ratio reflects ongoing biotransformation. Because O-demethylenation is CYP2D6-dependent, pharmacogenetic variability and co-medication with selective serotonin reuptake inhibitors (SSRIs) can delay MDA formation and complicate interpretation. Therefore, ratio-based inference should always be contextualized with patient factors and, when clinically or forensically relevant, confirmed by LC-MS/MS [130,131].

In most clinical and regulatory contexts, initial drug detection continues to rely on immunoassays targeting parent compounds or primary metabolites. While these assays are widely accessible and offer rapid results, they are prone to cross-reactivity with structurally related substances, potentially leading to false-positive outcomes. Therefore, any discordant or clinically significant findings should be verified using chromatographic techniques coupled with mass spectrometry. High-resolution LC-MS/MS now enables broad, simultaneous coverage of many stimulants of this class and their metabolites across matrices [46,131]. Specimen validity assessment is essential and includes creatinine concentration, specific gravity, urinary pH, and screening for oxidizing adulterants. As discussed in the pharmacokinetics section, urinary pH markedly affects amphetamine excretion and MA/AM ratios, and should therefore be considered in interpretation. Any suspicion of adulteration should prompt recollection and targeted LC-MS/MS confirmation [46,131]. Individuals attempting to evade detection often resort to urine dilution, ingestion of masking agents or oxidizing adulterants, or even substitution with drug-free samples. In response, laboratories apply specimen validity testing, including creatinine concentration, specific gravity, and oxidant checks, to identify tampering. These safeguards are essential to maintain the integrity of ATS screening in both clinical and forensic settings [132]. In routine practice, bupropion and its metabolites are among the most common causes of apparent “amphetamine” reactivity, and aripiprazole has produced positive screens in children and adolescents, both examples underline the need to confirm before drawing conclusions [133].

Recent metabolomics studies of amphetamine-type stimulants, including methamphetamine and its analogues, employ nuclear magnetic resonance spectroscopy (NMR) and LC-MS in untargeted metabolomic profiling, coupled with supervised machine-learning algorithms such as random forest, support vector machines, and artificial neural networks. These approaches identify metabolic pathway perturbations associated with different phases of use or withdrawal and generate biomarker panels that can predict exposure history and time-since-use. The combination of metabolomics and machine learning thus provides both mechanistic insight and potential forensic applications [134].

### 4.1. Screening and Cross-Reactivity in Clinical and Forensic Settings

Screening in clinical and workplace settings typically uses amphetamine-calibrated immunoassays with thresholds of 500 to 1000 ng/mL expressed as amphetamine equivalents, although United States federal programs use 500 ng/mL for screening and 250 ng/mL for confirmation [46,132]. In Europe, the European Workplace Drug Testing Society (EWDTS) recommends a screening cut-off of 500 ng/mL and confirmation at 250 ng/mL for amphetamine-class stimulants, harmonizing with but not identical to U.S. SAMHSA guidelines [135]. Cross-reactivity is clinically significant, with common interfering substances including bupropion, which is frequently encountered in emergency and outpatient settings, and aripiprazole, particularly relevant in pediatric and adolescent toxicological screens. Additional interferents include pseudoephedrine, phentermine, atomoxetine, metoprolol, the labetalol metabolite 1-methyl-3-phenylpropylamine, and fluoroquinolone antibiotics such as ofloxacin and moxifloxacin [134,136,137]. Discordant or consequential results should be confirmed by gas or liquid chromatography coupled to mass spectrometry to ensure analytical specificity [46,138]. Clinical pearls for ATS testing include several interpretive caveats of practical relevance. Apparent “amphetamine” positivity due to cross-reactants such as bupropion or aripiprazole should always be confirmed by LC-MS/MS before clinical or forensic conclusions are drawn. Conversely, a negative amphetamine-class screen in the setting of suspected MDMA or MDEA use necessitates a targeted LC-MS/MS panel to avoid false reassurance. Finally, detection of a predominant L-methamphetamine enantiomer, particularly when the methamphetamine/amphetamine ratio appears atypical, should raise suspicion for selegiline therapy or over-the-counter decongestant exposure [134]. A summary of common cut-off values, interfering substances, and interpretive caveats is presented in Table 3.

### 4.2. Metabolic Overview of Amphetamine—Type Stimulants

Across ATS, hepatic Phase I reactions like oxidative deamination, ring and side-chain hydroxylation, and N- or O-dealkylation, occur predominantly via CYP2D6 with contributions from CYP1A2 and CYP3A4, and are followed by Phase II conjugation, mainly glucuronidation and sulfation. Methamphetamine undergoes CYP2D6 mediated N-demethylation to amphetamine, creating intrinsic overlap between parent and metabolite. 3,4-Methylenedioxymethamphetamine is O-demethylenated to 3,4-methylenedioxyamphetamine and further hydroxylated to HMMA before conjugation. Interindividual CYP2D6 variability and co-medication may modulate these patterns. The N-ethyl analogue MDEA yields DHEA and HMEA, with minor N-deethylation to MDA; detection of HMEA supports recent MDEA exposure. By contrast, finding PMA or PMMA reflects non-standard synthesis or adulteration rather than endogenous metabolism and has specific forensic value. These overlapping pathways account for cross-detection between parent compounds and metabolites and necessitate that ratio-based findings, supplemented where appropriate by chiral data, be interpreted in clinical context and confirmed by mass spectrometry [114,130,131,140,141,142]. The pharmacokinetic characteristics of the main amphetamine-type stimulants are summarized in Table 4 [109,110,114,130,131,132,133,134,135,136,137,138,139,140,141,142,143,144,145,146,147,148,149,150,151,152,153,154,155].

### 4.3. Methamphetamine: Pharmacokinetics and Metabolism

Methamphetamine is efficiently absorbed by oral, intranasal, or intravenous routes, readily penetrates the central nervous system, and typically has a longer duration of action than amphetamine. Biotransformation is dominated by CYP2D6 mediated N-demethylation to amphetamine, with additional para-hydroxylation to 4-hydroxymethamphetamine and 4-hydroxyamphetamine followed by conjugation. Renal elimination is strongly pH-dependent: acidic urine markedly increases excretion of unchanged drug, whereas alkaline urine reduces clearance. For reporting and interpretation, urinary pH should be documented; acidic urine increases the renal elimination of unchanged amphetamine and methamphetamine, whereas alkaline urine reduces it and can inflate MA/AM ratios. For timing and source attribution, the urinary AM/MA ratio generally rises with time after exposure and should be interpreted with urine pH, dose, and clinical context, when attribution remains uncertain, enantioselective testing can distinguish therapeutic or over-the-counter L-methamphetamine exposure from illicit or prescription D-methamphetamine sources [114,130,139]. A simplified scheme of methamphetamine metabolism is shown in Figure 5, while key interpretive considerations regarding methamphetamine versus amphetamine intake are summarized in Table 5.

Key interpretive considerations include high methamphetamine with moderate amphetamine (MA/AM >2–3 in early hours) is consistent with methamphetamine intake. Conversely, high amphetamine with little or no methamphetamine suggests primary amphetamine use or a late collection window. Urinary pH also modifies these ratios: alkalinization prolongs MA half-life and raises ratios, whereas acidification accelerates clearance. When attribution remains uncertain, chiral (D/L) analysis is recommended [114,130]. Thus, methamphetamine interpretation requires careful integration of pharmacokinetics, urine pH, and enantiomeric profiling to avoid misclassification of use.

### 4.4. MDMA and MDA: Parent-Metabolite Relationships

For 3,4-methylenedioxymethamphetamine, the relation between the parent drug and 3,4-methylenedioxyamphetamine informs recency: higher MDMA/MDA ratio suggests recent ingestion, whereas a rising MDA/MDMA ratio reflects ongoing biotransformation. Given the CYP2D6-dependent O-demethylenation and frequent co-medication with selective serotonin reuptake inhibitors, ratio-based inference should account for pharmacogenetics and drug–drug interactions and be confirmed by liquid chromatography-mass spectrometry when results are consequential [131,144]. Taken together, parent-metabolite ratios provide valuable timing information but require confirmatory testing to ensure accurate attribution of MDMA versus MDA intake. An overview of MDMA, its metabolites, and their interpretive significance is provided in Table 6.

### 4.5. MDEA: Metabolic Pathways and Biomarkers

3,4-Methylenedioxyethylamphetamine is the N-ethyl analogue of MDMA. In humans it undergoes O-demethylenation to 3,4-dihydroxyethylamphetamine (DHEA), followed by O-methylation to 4-hydroxy-3-methoxyethylamphetamine (HMEA) and conjugation; a minor N-deethylation pathway yields MDA. Detection of HMEA in urine or blood supports recent MDEA exposure and complements MDMA: MDA-based timing. Confirmation by targeted liquid chromatography–mass spectrometry is recommended when source attribution is disputed [140,142]. Therefore, MDEA biomarkers, though informative, must always be interpreted in a broader metabolic and analytical context.

### 4.6. Synthetic Impurities and Adulterants of ATS

Detection of paramethoxyamphetamine or paramethoxymethamphetamine indicates non-standard synthesis or adulteration rather than endogenous metabolism and carries specific forensic and toxicological significance. In practice, simultaneous detection of PMA/PMMA, together with a D-enantiomeric predominance and atypical parent/metabolite ratios, provides compelling forensic evidence of illicit manufacture and is used in judicial and disciplinary investigations [141].

### 4.7. Synthetic Cathinones: Analytical and Toxicological Considerations

Routine immunoassays calibrated for amphetamines exhibit limited cross-reactivity with synthetic cathinones, often resulting in false-negative findings. Therefore, reliable detection and differentiation necessitate the use of targeted LC-MS/MS panels or, optimally, high-resolution mass spectrometry. Notably, since synthetic cathinones lack approved therapeutic indications, their identification unequivocally indicates illicit or designer-drug use, underscoring the critical importance of confirmatory mass spectrometric analysis for accurate forensic and clinical interpretation [149]. In practice, detection of synthetic cathinones is a direct marker of illicit exposure, making robust analytical confirmation crucial for both forensic and medical evaluations. Representative synthetic and natural cathinones with their main metabolites and analytical challenges are listed in Table 7.

### 4.8. Chiral (Enantiomeric) Analysis for Source Attribution

When patient history and initial screening results are inconsistent, chiral analysis provides decisive clarification. A predominance of the L-enantiomer of methamphetamine supports exposure from selegiline metabolism or inhaled decongestants, whereas enrichment of the D-enantiomer indicates illicit use or prescription D-methamphetamine. Lisdexamfetamine yields only D-amphetamine in vivo, so detection of D-amphetamine alone supports therapeutic adherence. Chiral findings should be interpreted together with the time dependent AM/MA ratio and urine pH [46,138]. Ultimately, enantiomeric profiling serves as a decisive tool for distinguishing legitimate medical therapy from recreational or illicit ATS use. Practical interpretive clues from enantiomeric analysis are summarized in Table 8.

In conclusion, the interpretation of amphetamine-type stimulant biomarkers requires a multidimensional approach that integrates broad screening with confirmatory mass spectrometry. Parent-metabolite ratios, chiral (enantiomeric) profiles, and impurity markers provide complementary layers of information, but their clinical value emerges only when considered alongside pharmacokinetic factors, urine pH, dosing context, and patient history. Ultimately, refining these analytical strategies is not only essential for forensic accuracy but also for improving clinical management, tailoring therapeutic monitoring, and guiding public health responses to the rising global burden of ATS misuse. Aligning such approaches with those already established for other controlled substances, such as opioids, will further strengthen diagnostic accuracy and enhance both therapeutic and public health outcomes. Future perspectives should focus on standardizing cut-off values, validating novel biomarkers, and further integrating metabolite-based strategies into clinical and forensic practice.

## 5. Benzodiazepines and Z-Drugs

Benzodiazepines are a class of psychoactive drugs that were originally introduced for therapeutic purposes in the mid-20th century. Since then, benzodiazepines have been widely used for their sedative-hypnotic, anxiolytic, anticonvulsant, and muscle relaxant properties [156,157]. They act as positive allosteric modulators at the γ-aminobutyric acid type A (GABAA) receptor. The endogenous ligand for this receptor is γ-aminobutyric acid (GABA), the primary inhibitory neurotransmitter in the central nervous system (CNS). Benzodiazepines enhance the affinity of GABA for its receptor, increasing chloride ion influx and promoting neuronal hyperpolarization, thereby leading to a general CNS depressant effect [157]. Clinically, they are commonly prescribed for the acute management of anxiety disorders, insomnia, alcohol withdrawal, status epilepticus, and as premedication for medical or surgical procedures [156,158]. Benzodiazepines have a relatively high therapeutic index, particularly when compared to older barbiturates, contributed to their widespread adoption in medical practice [159,160]. However, the long-term use of benzodiazepines is constrained by the potential for tolerance, dependence, and withdrawal phenomena [158,160,161]. Chronic administration leads to neuroadaptive changes at the GABA_A_ receptor complex, diminishing therapeutic efficacy and predisposing individuals to withdrawal symptoms upon abrupt cessation. These symptoms can range from rebound anxiety and insomnia to more severe manifestations such as seizures and psychosis [161]. Moreover, benzodiazepines possess a significant potential for misuse, especially when used in conjunction with other CNS depressants such as opioids or alcohol, which can potentiate respiratory depression and increase overdose risk [158,162].

Z-drugs, including zolpidem, zopiclone, and zaleplon, are non-benzodiazepine hypnotics primarily prescribed for short-term treatment of insomnia [163,164]. They act selectively on the benzodiazepine binding site of the GABA_A_ receptor, producing sedative-hypnotic effects with minimal anxiolytic or muscle-relaxant activity compared to classical benzodiazepines [163,165]. Although initially considered safer alternatives due to lower risk of dependence, accumulating evidence indicates that Z-drugs can still lead to tolerance, withdrawal, and misuse, particularly in patients with a history of substance use disorders [164,166]. Adverse events such as cognitive impairment, falls, and complex sleep-related behaviors have also been reported, especially among elderly populations [165,166].

From a public health perspective, the non-medical use of benzodiazepines and Z-drugs has become an increasing concern, often associated with polydrug abuse and adverse outcomes as well as an implication in drug-facilitated crimes [160,162]. Therefore, current prescribing guidelines advocate for the lowest effective dose over the shortest duration possible, along with regular monitoring of efficacy and safety as well as periodic reassessment of ongoing need [161,163,164]. In addition to the abuse potential of benzodiazepines, they are well-known for their use in DFSA [167].

Therefore, despite benzodiazepines remain an important pharmacological tool in acute settings, their risk-benefit ratio necessitates careful patient selection and monitoring to mitigate the likelihood of misuse [160,161]. In this regard, data from the National Survey on Drug Use and Health from the United States indicated that nearly 17.7% of all individuals who used benzodiazepines and 9.2% of those who took Z-drugs reported misuse within the past year [168]. Data from European Union Drugs Agency (EUDA) indicate that benzodiazepines are frequently implicated in drug-induced deaths across Europe, especially in drug users who also take opioids and other central nervous system depressants. Moreover, designer benzodiazepines such as etizolam have become an increasing public health and forensic toxicology concern due to their growing availability through illicit online markets and the absence of pharmaceutical regulation. They often exhibit high potency, rapid onset, and unpredictable pharmacokinetics, leading to a high risk of overdose and dependence. Their metabolism is poorly characterized compared to traditional benzodiazepines, with recent metabolomic studies identifying numerous phase I and phase II metabolites. This metabolic diversity complicates toxicological interpretation and detection, as parent compounds are often absent in biological samples, while metabolites lack validated reference standards [169,170].

Benzodiazepines encompass a range of compounds and are characterized with varying pharmacokinetic profiles, including differences in onset and duration of action as well as metabolic pathways which impact their clinical use [156,159]. The main clinically used benzodiazepines and Z-drugs are listed in Table 9 along with their pharmacokinetic properties and major metabolites.

### 5.1. Toxicological Analysis of Benzodiazepines and Z-Drugs

Several methods for laboratory detection of benzodiazepines and Z-drugs are available. A traditional method is the immunoassay-based analysis, which targets either the parent compound or common metabolites [171,199]. Although these assays are widely accessible and cost-effective, they exhibit variable sensitivity and a considerable risk of cross-reactivity among structurally similar benzodiazepines, leading to potential false positives or negatives [167]. To overcome these limitations, confirmatory analytical methods such as GC–MS and LC–MS/MS are employed as the gold standard for definitive identification and quantification [176,200]. These chromatographic techniques provide superior specificity, sensitivity, and quantitative accuracy, enabling simultaneous detection of multiple benzodiazepines, Z-drugs, and their metabolites in complex biological matrices such as plasma, urine, hair, and oral fluid. Recent advances include HRMS and targeted metabolomic profiling, which have markedly improved the discrimination between therapeutic use and misuse through detailed metabolic signatures [171,201].

Recent advances in analytical toxicology are microsampling, particularly dried blood spot (DBS) and volumetric absorptive microsampling (VAMS) techniques, which offer minimally invasive, stable, and logistically convenient alternatives for assessing plasma-equivalent concentrations of both parent drugs and metabolites [202]. These approaches facilitate therapeutic drug monitoring and forensic assessment even in resource-limited or field conditions, without compromising analytical accuracy. Furthermore, hair segmental analysis has gained traction for its ability to provide a retrospective timeline of drug consumption over weeks or months, offering insights into long-term use or accidental intake [203,204]. Recent data emphasizes systematic metabolic characterization using LC-HRMS combined with computational metabolomics and machine-learning-assisted spectral annotation for benzodiazepines’ analysis, particularly for emerging designer or novel benzodiazepines. These approaches facilitate identification of unexpected or low-abundance metabolites and are critical when authentic analytical standards are unavailable. Current reviews highlight the integration of in silico metabolic prediction tools (for example, BioTransformer or GLORYx) with experimental mass-spectrometric data to rapidly expand forensic toxicology panels [205].

Plasma and serum remain the major biological matrices for assessing recent use and pharmacokinetic parameters as they closely reflect the pharmacologically active fraction of the drug. However, due to short detection windows from typically a few hours to days, these matrices may not reliably differentiate between therapeutic use and sporadic misuse, particularly for short-acting agents such as midazolam or zolpidem [206,207]. In contrast, urine testing offers broader detection windows and is ideal for screening and compliance monitoring as it captures both parent compounds and conjugated metabolites. Reported confirmatory cut-offs for benzodiazepines and Z-drugs depend on matrix and method. Additionally, interpretive challenges arise because urinary concentrations are influenced by hydration status, metabolism, and renal clearance, making parent/metabolite ratios less consistent across individuals [46,208]. Hair analysis provides a unique retrospective timeline, allowing assessment of chronic versus isolated use over several weeks or months [209]. Saliva testing has recently gained interest as a non-invasive alternative that reflects the unbound, pharmacologically active fraction of benzodiazepines. Yet, the typically low drug concentrations and lipophilic nature of many benzodiazepines limit sensitivity and quantification reliability [210]. Finally, multi-matrix strategies, combining plasma or urine with hair or oral fluid, provide complementary insights, improving the ability to discriminate legitimate medical exposure from illicit or non-compliant use [211]. Table 10 summarize the available information about appropriate matrices, detection method and typical confirmatory cut-off values for major benzodiazepines and Z-drugs [170,171,172,173,174,175,176,177,178,179,180,181,182,183,184,185,186,187,188,189,190,191,192,193,194,195,196,197,198,199,200,201,202,203,204,205,206,207,208,209,210,211].

### 5.2. Interpretation of Analytical Findings

Interpreting analytical findings for benzodiazepines and Z-drugs remains a complex task due to extensive metabolic interconversion, variable pharmacokinetics, and individual differences in CYP450 enzyme activity. Many benzodiazepines such as diazepam, nordiazepam, temazepam, and oxazepam share a common metabolic cascade complicating the attribution of results to a specific parent compound. Therefore, analysis of parent-to-metabolite ratios is crucial to estimate time since exposure or differentiate therapeutic use from abuse [212,213].

#### 5.2.1. Diazepam

Diazepam is among the benzodiazepines frequently implicated in non-medical use and forensic detections, reflecting both its wide availability and frequent prescription [214,215]. Diazepam is a long-acting benzodiazepine owing to its prolonged elimination half-life as well as the formation of long-lived active metabolites (N-desmethyldiazepam/nordiazepam, temazepam and oxazepam), which prolong clinical and toxicological effects and extend detection windows. Diazepam is extensively metabolized in the liver predominantly by cytochrome P450 enzymes, notably CYP2C19 and CYP3A4, which mediate N-demethylation and other reactions to form active metabolites. Interindividual variability in these enzymes contributes to differences in clearance and exposure [170,216,217,218]. Importantly, several metabolites (e.g., temazepam, oxazepam, and nordiazepam) are themselves clinically used benzodiazepines, which further complicates interpretation of toxicological findings because detection of a particular metabolite may reflect direct administration of that metabolite as a prescription drug rather than metabolism of diazepam [170,214,217]. The metabolic pathway of diazepam is presented in Figure 6.

After oral or parenteral diazepam administration, the parent drug is relatively rapidly absorbed and distributed while the main active metabolite nordiazepam accumulates and declines more slowly because of its longer elimination half-life. Consequently, the diazepam to nordiazepam concentration ratio falls with time after ingestion. That is confirmed by several controlled-dose pharmacokinetic studies and forensic case series. Higher ratio values are commonly observed soon after diazepam dosing whereas low ratios (metabolite-dominant profiles) are typical later or during chronic exposure [170,219]. In this regard, Wang et al. developed correlation models between diazepam/nordiazepam concertation ratio and time since oral diazepam ingestion and found that concentration-ratio based models predicted time since dose with reasonable accuracy (prediction errors < 20%). This supports the general principle that a relatively high diazepam to nordiazepam concentration ratio favors more recent intake [220].

The pattern and ratio of metabolites along with clinical data, offers a multidimensional framework for interpretation. The presence of all three metabolites (nordiazepam, temazepam and oxazepam) strongly supports diazepam use, particularly when accompanied by measurable diazepam. It indicates sufficient time for sequential metabolism through both N-demethylation and hydroxylation pathways. The relative proportions of nordiazepam and oxazepam reflect the metabolic progression and may also aid in approximating time since ingestion [170,213]. Isolated nordiazepam presence cannot confirm diazepam use without supporting evidence (e.g., parent diazepam or co-occurrence of minor metabolites). A sample containing only nordiazepam, especially at steady-state concentrations, may represent therapeutic use of nordiazepam rather than diazepam ingestion [175]. The presence of temazepam and/or oxazepam without diazepam or nordiazepam may be interpreted in several ways. Temazepam may originate from diazepam, but also from primary therapeutic administration. Oxazepam is the final common metabolite of diazepam, chlordiazepoxide, nordiazepam and temazepam, and is non-specific indicator for the parent compound. Detection of oxazepam alone typically suggests remote exposure or metabolism of another benzodiazepine, especially if diazepam and nordiazepam are absent or below quantitation limits [209]. It should be noted, that genetic polymorphism in CYP2C19 (poor metabolizers) or concurrent CYP inhibitors intake (e.g., omeprazole, fluoxetine) may change diazepam pharmacokinetics yielding higher diazepam and lower nordiazepam concentrations than expected for the time since dose, which could be mistaken for recent ingestion if not contextualized [216]. Although the diazepam to nordiazepam concentration ratio is informative, the literature does not support a universally valid numeric cut-off that applies across matrices and populations. Ratio values are influenced by dose, formulation, time since dose, co-medications, hepatic and renal function, age, and analytical variability. Therefore, the designation of specific threshold values of diazepam/nordiazepam ratio indicating recent and chronic use should be avoided in formal reporting unless supported by validated, lab-specific models [213,219].

#### 5.2.2. Other Benzodiazepines

The interpretation of analytical results for alprazolam and other short- to intermediate-acting benzodiazepines also requires careful consideration of metabolic pathways and analytical limitations. Alprazolam is characterized with a relatively short half-life and rapid addiction [221]. It undergoes predominant oxidation via CYP3A4, yielding α-hydroxyalprazolam and 4-hydroxyalprazolam as its principal metabolites that are further conjugated with glucuronic acid and excreted through the kidney. One of the metabolites, α-hydroxyalprazolam is pharmacologically active and is commonly detected in biological samples [200,212,216]. The parent-to-metabolite ratio may offer qualitative insight into time since ingestion as recent exposure typically yields detectable parent alprazolam with lower α-hydroxyalprazolam, whereas higher levels of the metabolite suggests delayed or past intake [171].

Bromazepam presents a distinct interpretive challenge due to partial metabolism to hydroxybromazepam and minor desmethyl derivatives, which exhibit limited specificity for source identification. The presence of both parent compound and its hydroxy-metabolite in plasma or urine suggests recent exposure, whereas isolated hydroxybromazepam, especially in urine should be interpreted cautiously, as interindividual glucuronidation capacity significantly alters excretion kinetics [68,175].

Flunitrazepam undergoes metabolic processes of reduction, hydroxylation and N-demethylation yielding 7-aminoflunitrazepam, 3-hydroxyflunitrazepam and N-desmethylflunitrazepam, respectively. The major metabolite, 7-aminoflunitrazepam remains a key biomarker for both therapeutic and illicit use. The detection of 7-aminoflunitrazepam without parent compound indicates prior ingestion, while coexistence of unchanged flunitrazepam implies recent use. Because of its metabolic stability 7-aminoflunitrazepam persists in urine longer than other benzodiazepine metabolites, extending interpretive windows up to several days. It should be noted that genetic polymorphism, drug–drug interactions as well as stability of flunitrazepam during storage may result in large interindividual variability in the toxicological results [222,223]. Additionally, in case of suspicion of a crime, flunitrazepam and 7-aminoflunitrazepam may be detected in hair samples more than 6 months later [221]. Moreover, knowing the concentration of flunitrazepam and 7-aminoflunitrazepam as well as its parent-to-metabolite concentration ratios in hair samples may be useful for the differentiation of therapeutic drug intake from random ingestion [224].

#### 5.2.3. Z-Drugs

In contrast to benzodiazepines, Z-drugs have short half-lives and undergo rapid hepatic biotransformation. Zolpidem and zopiclone/eszopiclone are predominantly metabolised by CYP3A4 while zaleplon is metabolized primarily by the enzyme aldehyde oxidase into inactive metabolites. Zopiclone/eszopiclone has an active metabolite, (S)-desmethylzopiclone. The detection window for Z-drugs in urine is longer than in blood/plasma (approximately 24–48 h) [201]. For zopiclone and eszopiclone, metabolite-pattern analysis rather than parent-only detection should be used since unchanged parent drug is excreted in low amounts and major metabolites predominate in biological fluids [225]. Furthermore, hair samples analysis have identified enantiomeric differences in zopiclone metabolites that may be useful in distinguishing legal therapeutic formulations from illicit analogues [226].

In conclusion, changes in parent-to-metabolite ratio with time is informative in the context of misuse detection. However, variability in CYP450 activity, drug–drug interactions with CYP450 inhibitors or inducers as well as dosage form used may affect these ratios [156]. Additionally, existing ratio-vs-time models are promising but typically derived from small controlled cohorts or forensic case series. Additionally, population variability remains substantial so metabolite profiling alone cannot definitively attribute drug origin without clinical correlation or prescription verification. Given these limitations, combined evaluation of metabolite pattern, relative ratios, and biologic matrix type remains the most robust approach for distinguishing diazepam misuse from legitimate use [46,170,213].

## 6. Cannabinoids

Cannabinoids are chemical compounds that exert their effects by binding to cannabinoid receptors. Endogenous cannabinoids are lipid-based, diffusible, short-lived neuromodulators that transmit signals retrogradely from depolarized postsynaptic neurons to presynaptic nerve terminals, thereby inhibiting neurotransmitter release [227]. The best-characterized endocannabinoids are arachidonoyl ethanolamide (anandamide) and 2-arachidonoyl glycerol. They are synthesized on demand from precursors in the neuronal cell membrane [228].

Phytocannabinoids are naturally produced by the *Cannabis sativa* L. plant. Over 100 phytocannabinoids have been identified so far, with Δ^9^-tetrahydrocannabinol (THC) and cannabidiol (CBD) being the most extensively studied [229]. While cannabis and its derivatives have been historically used for both medicinal and recreational purposes, modern pharmacological research has led to the development of selective cannabinoid receptor modulators [230].

### 6.1. Therapeutic Use of Cannabinoids

Cannabinoids approved by the FDA and EMA include the plant-derived CBD (Epidyolex^®^) and the synthetic cannabis-related drug products dronabinol (Marinol^®^, Syndros^®^) and nabilone (Cesamet^®^) [231,232]. Nabiximols (Sativex^®^), a whole-plant extract containing both THC and CBD in a 1:1 ratio, has been approved by the EMA, but not by the FDA. Cannabis in its natural plant form is not approved for use by either the FDA or EMA. However, besides the centralized EMA procedure for marketing authorization, cannabinoid-based products may also be authorized through decentralized, mutual recognition, or national procedures of individual member states, which can limit their availability and use [232].

The effects of cannabinoids are primarily mediated through two G protein–coupled receptors: CB_1_ and CB_2_. CB_1_ receptors are abundant in the central nervous system, particularly in areas involved in pain, memory, emotion, and reward, while CB_2_ receptors are more prevalent in peripheral tissues, especially in immune cells [233]. Depending on the specific compound and receptor subtype, cannabinoids can act as full agonists, partial agonists, antagonists, or inverse agonists [230].

In clinical settings, cannabinoid-based therapies have demonstrated efficacy in managing chronic neuropathic pain, spasticity in multiple sclerosis, chemotherapy-induced nausea, and certain forms of epilepsy [233,234,235,236]. Table 11 presents the approved clinical uses of cannabinoids. However, their therapeutic potential is complicated by psychoactive effects, particularly those linked to CB_1_ activation. Adverse effects may include cognitive impairment, anxiety, tachycardia, and, in predisposed individuals, an increased risk of psychosis or mood disorders [237]. Tolerance and dependence may develop with long-term use, and withdrawal syndromes have been documented following cessation after repeated exposure to psychoactive cannabinoids [233]. Despite these challenges, the risk of fatal overdose is relatively lower compared to opioids and growing clinical evidence supports cautious use of cannabinoids as pharmacological agents, alongside public health and regulatory considerations [230,238].

### 6.2. Recreational Use of Cannabinoids

In response to growing psychiatric and public health concerns, the DSM-5 introduced Cannabis Use Disorder (CUD) as a distinct diagnosis within substance-related and addictive disorders. CUD involves a maladaptive pattern of cannabis use causing clinically significant impairment, characterized by tolerance, craving, withdrawal, and failed attempts to reduce consumption [18,239]. Epidemiological data from NESARC-III indicate a 12-month prevalence of 2.5% and a lifetime prevalence of 6.3% among U.S. adults [240], with an estimated 27% lifetime transition risk from cannabis use to CUD—particularly among men and early-onset users [241]. CUD frequently co-occurs with anxiety, depression, bipolar, and other substance use disorders [239,242,243]. Although less lethal than opioid dependence, CUD substantially contributes to disability-adjusted life years (DALYs), ranking among the leading drug-related causes in young populations [244].

Legalization trends and reduced risk perception may further increase misuse, especially among adolescents [245]. Commonly misused cannabinoids include THC, high-potency cannabis extracts, synthetic cannabinoids (“Spice,” “K2”), and, to a lesser extent, CBD [240]. Inhalation remains the predominant route, followed by oral ingestion. Up to one-third of U.S. adults who used cannabis in the past year met DSM-5 criteria for CUD [246]. Despite this, no pharmacological treatments are approved, and management relies on psychosocial interventions such as cognitive-behavioral therapy, motivational enhancement therapy, and contingency management [247]. Treatment uptake remains very low—only 4.7% of adults with CUD reported receiving any form of care in 2022 [248].

A meta-analysis involving over 23,000 regular users found that 47% experience withdrawal symptoms such as irritability, anxiety, insomnia, and appetite loss, which often contribute to relapse [247]. These findings underscore the urgent need for improved biomonitoring, development of pharmacotherapies, and broader access to evidence-based interventions to mitigate the rising burden of cannabis-related disorders.

### 6.3. Toxicological Analysis of Cannabinoids

Toxicochemical analysis of cannabinoids has become essential not only in forensic and toxicological contexts but also in clinical, occupational, and regulatory settings. Distinguishing therapeutic or controlled cannabinoid use from illicit consumption is increasingly important for medical cannabis programs, workplace policies, and public health surveillance, underscoring the need for robust and reliable detection methodologies. Laboratory detection of cannabinoids commonly begins with immunoassay-based screening, which targets either parent compounds (e.g., THC) or major metabolites (e.g., THC-COOH). These assays are valued for their speed, automation, and cost-effectiveness, but they are inherently susceptible to cross-reactivity with structurally similar cannabinoids or interfering compounds (e.g., therapeutic cannabinoids or analogs). To mitigate false positives or negatives, confirmatory tests are deployed, relying on chromatographic separation coupled with mass spectrometry, which remains the analytical gold standard.

Recent developments in cannabinoid toxicochemistry include LC-MS/MS methods capable of simultaneous identification of multiple phytocannabinoids and metabolites directly from urine samples (e.g., nine analytes including THC, 11-OH-THC, THC-COOH, cannabidiol, etc.) without extensive sample preparation [249]. Similarly, methods for detecting large panels of synthetic cannabinoid metabolites (up to 61 analytes) via solid-phase extraction and LC-MS/MS have been validated with detection limits in the low ng/mL range [250]. In another advance, quantitative assays have been optimized to resolve interferences between Δ^8^-THC, Δ^9^-THC, and their metabolites, with lower limits of quantitation of 10 ng/mL [251]. These analytical innovations improve the sensitivity, specificity, and throughput of cannabinoid testing. They also enable differentiation between illicit cannabis use and pharmaceutical cannabinoid therapies, such as dronabinol, by distinct metabolite patterns [252].

Recent work applies untargeted mass spectrometry–based metabolomics, MS/MS spectral networking, and machine learning models to map the chemical diversity of both phytocannabinoids (plant-derived compounds such as THC and CBD and synthetic cannabinoids, as well as their human metabolites. Large in vitro (cellular or microsomal) and in vivo (animal or clinical) metabolic profiling studies have identified robust urinary marker metabolites that enable differentiation between various cannabinoid classes. Combining experimental human liver microsome data with machine-learning-driven spectral mining allows prioritization of the most probable metabolites for confirmatory testing and quantitative analysis [253].

The evolving landscape of cannabinoid legalization and diversified product formulations underscores the importance of precise monitoring in clinical and regulatory settings. Table 12 summarizes the main methods for detection of cannabinoids and their metabolites in blood and urine samples.

### 6.4. Cannabinoids Metabolism

In addition to mastery of sensitive analytical methods, a deep understanding of cannabinoid metabolism is essential when interpreting biological samples such as blood or urine. Many clinically relevant cannabinoids (or their precursors) are biotransformed into active or inactive metabolites, and in some cases metabolites or conjugates might be mistaken for contaminants or impurities if metabolic pathways are disregarded. Overlooking these transformations can lead to misinterpretation of exposure, use patterns, or drug interactions.

Cannabinoid metabolism occurs predominantly in the liver, proceeding through two general phases. Phase I reactions include oxidative and reductive transformations, largely mediated by cytochrome P450 (CYP) enzymes, while Phase II reactions typically involve conjugation (e.g., glucuronidation) to increase water solubility and promote excretion. For many phytocannabinoids, the principal CYP enzymes implicated are CYP2C9, CYP3A4, and to a lesser extent CYP2C19 [259]. After Phase I oxidation, metabolites often undergo glucuronidation via UDP-glucuronosyltransferases (UGTs) [260,261]. A prototypical example is Δ^9^-THC metabolism. After hepatic uptake, THC is oxidized by CYP2C9/CYP3A4 to 11-hydroxy-THC (11-OH-THC), an active metabolite, which may itself be further oxidized to 11-nor-9-carboxy-THC (THC-COOH), an inactive metabolite [261,262,263,264]. The THC-COOH is then conjugated (often via UGT1A1, UGT1A3) to form a glucuronide, which is the principal urinary excretion product [260,261]. In human hepatic microsomes, Δ^8^-THC is also primarily oxidized to 7α-hydroxy-Δ^8^-THC and 11-hydroxy-Δ^8^-THC [265]. The pharmacokinetic variability in cannabinoid metabolism arises from multiple sources. Genetic polymorphisms in CYP2C9 (“poor, intermediate, or ultra-rapid metabolizers”) may substantially affect THC clearance and active metabolite formation [264]. In addition, cannabinoids and their metabolites themselves can act as inhibitors (or sometimes inducers) of CYP and UGT enzymes, creating potential drug–drug interactions. For instance, 11-OH-THC and THC-COOH-glucuronide have been shown to competitively inhibit CYP2B6, CYP2C9, and CYP2D6 [266]. Moreover, CBD demonstrates potent inhibition of several UGTs (e.g., UGT1A9, UGT2B7), which may affect the conjugation of coadministered drugs [267]. Unlike opioids that might occasionally generate confusion through minor side metabolites, cannabinoid metabolic pathways generally yield predictable oxidation-plus-conjugation cascades. However, analytical challenges remain: metabolites may accumulate at concentrations exceeding the parent compound in plasma, and the lipophilicity, binding to tissues, and re-distribution can further complicate interpretation. Thus, correctly attributing signals in biological matrices to parent cannabinoids or their metabolites demands careful method validation and metabolic context. Table 13 outline the pharmacokinetic properties and metabolic pathways of the best-studied cannabinoids [260,261,262,263,264,265,266,267,268,269,270,271,272].

### 6.5. Differentiating Therapeutic Epidyolex^®^ Use from Recreational Cannabis Consumption

Discriminating between the medically supervised use of Epidyolex (a purified oral CBD solution) and the non-medical use of cannabis products represents a growing challenge in clinical and forensic toxicology. Although both derive from *Cannabis sativa* L., the chemical composition, pharmacokinetic profiles, and patterns of use differ significantly between regulated CBD pharmaceuticals and recreational cannabis, which often contains high levels of THC and variable amounts of CBD [273].

Epidyolex^®^ is administered orally in precise, standardized doses, leading to gradual absorption and relatively stable plasma CBD concentrations. Peak CBD levels typically occur within 2–5 h, with prolonged half-life and limited psychoactivity [274]. In contrast, recreational cannabis is commonly smoked or vaporized, delivering rapid systemic THC absorption with psychoactive effects appearing within minutes. This route results in a shorter detection window for cannabinoids in blood but often leaves distinct metabolic signatures. Unlike Epidyolex, recreational cannabis products contain psychoactive THC, often in high concentrations (10–30% by dry weight), and are rarely standardized, leading to unpredictable pharmacological and toxicological effects.

Epidyolex^®^ is composed of >98% pure CBD with only trace or non-detectable levels of THC (<0.2%, as per regulatory requirements). Its use results in significant plasma and urinary concentrations of CBD and its metabolites, including 7-hydroxy-CBD (7-OH-CBD) and 7-carboxy-CBD (7-COOH-CBD) [275]. Importantly, THC and its metabolites (e.g., 11-OH-THC, THC-COOH) are absent or present only at trace levels, typically below detection thresholds in compliant therapeutic use. In contrast, recreational cannabis leads to dominant THC metabolite profiles, with high levels of THC-COOH and low or inconsistent CBD metabolites, reflecting the THC-rich composition of most illicit cannabis. Moreover, the presence of minor cannabinoids such as CBG, CBC, and THCV, more prevalent in whole-plant cannabis products, further supports a recreational origin.

The differentiation between Epidyolex and recreational cannabis exposure relies on advanced analytical methodologies, particularly LC–MS/MS, enabling simultaneous quantification of CBD, THC, their metabolites, and minor cannabinoids at ng/mL sensitivity. Key discriminators include:CBD/THC ratio: Epidyolex usage shows high CBD with negligible THC, while recreational use presents THC dominance.Presence of 7-OH-CBD and 7-COOH-CBD: Indicative of pharmaceutical CBD metabolism.Absence of plant-derived impurities or minor cannabinoids in Epidyolex, versus their variable presence in unregulated cannabis.

Enantioselective and isotope ratio mass spectrometry may further distinguish synthetic or purified cannabinoids from plant extracts, though such techniques are not routinely applied in most forensic settings.

Interpretation of cannabinoid profiles must be contextualized within the patient’s medical history, prescription records, and timing of sample collection. Epidyolex use can be objectively verified via consistent CBD metabolite patterns, absence of significant THC exposure, and documentation of treatment. In contrast, THC-dominant profiles, inconsistent CBD presence, and detection of non-pharmaceutical cannabinoids support non-medical cannabis use. As CBD-based medications gain broader approval for conditions such as Dravet syndrome and Lennox-Gastaut syndrome, precise differentiation becomes essential in clinical monitoring, legal adjudication, and occupational or driving-related drug testing. Reliable discrimination ensures accurate assessment of compliance, protects patients from misinterpretation of therapeutic use, and supports public health and safety standards.

### 6.6. Differentiating Therapeutic Dronabinol Use from Recreational Cannabis Consumption

The differentiation between therapeutic administration of dronabinol (synthetic Δ^9^-THC) and recreational use of natural cannabis containing THC poses significant analytical challenges in clinical and forensic toxicology. Despite dronabinol and natural THC being chemically identical, several pharmacokinetic, metabolic, and contextual factors enable distinction between these two sources.

Dronabinol is typically administered orally in standardized doses, resulting in a predictable pharmacokinetic profile characterized by relatively slow absorption, peak plasma concentrations reached approximately 2–4 h post-dose, and significant first-pass metabolism in the liver. In contrast, natural THC consumption commonly occurs via inhalation (smoking or vaporization), producing rapid absorption and peak plasma levels within minutes, accompanied by a shorter duration of detectable THC in biological matrices [262].

The metabolic pathways of synthetic dronabinol and natural THC substantially overlap, involving cytochrome P450 enzymes (primarily CYP2C9, CYP2C19, and CYP3A4) to generate active 11-hydroxy-THC (11-OH-THC) and subsequently inactive 11-nor-9-carboxy-THC (THC-COOH), as presented on Table 2. However, distinguishing markers may arise from metabolites of accompanying cannabis plant constituents absent in dronabinol formulations. For example, natural cannabis contains CBD and other minor cannabinoids, which metabolize via distinct pathways and can be detected alongside THC metabolites [276]. The absence of such cannabinoids and their metabolites in biological samples suggests pharmaceutical dronabinol ingestion.

High-sensitivity chromatographic techniques combined with mass spectrometry (e.g., LC-MS/MS) are widely used to detect and quantify THC, 11-OH-THC, THC-COOH, and minor cannabinoids in plasma, urine, and oral fluid. The detection of additional cannabinoids such as CBD or CBN alongside THC metabolites often suggests plant-derived cannabis use rather than pure dronabinol [252,277,278]. Emerging analytical approaches—including enantiomeric differentiation and isotope ratio mass spectrometry—hold promise for distinguishing synthetic versus plant-derived cannabinoids, though they remain mostly in the realm of advanced or specialized applications.

In forensic settings, contextual information such as prescription records and sample timing is routinely employed to complement analytical data when interpreting cannabinoid exposure. Controlled dronabinol administration typically yields more predictable plasma THC and metabolite profiles consistent with therapeutic dosing, whereas recreational cannabis use often presents with greater variability in concentrations and the co-detection of additional cannabinoids [262].

### 6.7. Differentiating Therapeutic Nabilone Use from Recreational Cannabis Consumption

Discriminating between the medically supervised use of nabilone (a synthetic cannabinoid and structural analogue of THC) and non-medical consumption of cannabis products is an important but often overlooked challenge in clinical and forensic toxicology. While both compounds act on the endocannabinoid system and share pharmacodynamic similarities, their chemical structure, route of administration, metabolic profiles, and origin differ substantially. This distinction is critical in contexts such as drug testing, forensic evaluations, and clinical compliance monitoring.

Nabilone is administered orally in capsule form under medical supervision, typically for chemotherapy-induced nausea and vomiting that is unresponsive to conventional antiemetics. Following oral administration, nabilone exhibits delayed absorption, with peak plasma levels occurring within 1–4 h, and a half-life of up to 35 h due to extensive tissue distribution and enterohepatic recirculation [278,279]. In contrast, recreational cannabis is commonly smoked or vaporized, resulting in rapid THC absorption, early peak concentrations, and shorter duration of systemic detection. This difference in route leads to distinct pharmacokinetic signatures: nabilone yields a slower onset and longer duration, while inhaled THC is fast-acting and short-lived.

Unlike cannabis, which contains a complex mixture of phytocannabinoids (THC, CBD, CBG, CBC, THCV, etc.), nabilone is a single synthetic compound, not found in nature. It does not contain cannabidiol (CBD) or other minor cannabinoids typically present in cannabis plant extracts.

Upon metabolism, nabilone produces unique synthetic metabolites that differ structurally from those of THC. Nabilone does not generate significant levels of THC-COOH or 11-OH-THC, the two primary metabolites of Δ^9^-THC commonly used to indicate cannabis use. Its metabolites include hydroxylated and carboxylated derivatives [279], but these are chemically distinct and require targeted analytical methods for confirmation. In contrast, recreational cannabis use leads to dominant THC metabolite profiles, typically with high concentrations of THC-COOH in urine and detectable 11-OH-THC in blood, often accompanied by low or absent CBD (depending on strain). The presence of minor cannabinoids such as CBG, CBC, and THCV further supports a plant-derived cannabis source.

Differentiating nabilone use from recreational cannabis consumption requires advanced confirmatory testing, as standard immunoassay drug screens may yield false positives due to cross-reactivity with THC metabolites. Key analytical approaches include the following:LC–MS/MS or GC–MS: Required for specific detection of nabilone and its unique metabolites, which are not identifiable through routine THC screening assays.Absence of THC-COOH and plant-based cannabinoids (CBD, CBN, CBG): Strongly indicative of synthetic cannabinoid use.Negative detection for minor cannabinoids: Supports pharmaceutical origin.Immunoassays: Prone to false positives due to structural similarity of nabilone and THC; not reliable alone for differentiation.

Some laboratories may also use chiral or isomer-specific analyses, though these are not widely implemented in routine testing.

Proper interpretation of cannabinoid results requires contextual information, including documented medical prescription for nabilone; patient history, indication for use, and dosing schedule; timing of sample collection relative to administration. In forensic and occupational settings, the detection of nabilone-specific metabolites in the absence of THC and its metabolites strongly supports legitimate therapeutic use. Conversely, the presence of THC-COOH, 11-OH-THC, and plant-derived cannabinoids indicates non-medical cannabis use. Because nabilone does not contain CBD or other minor phytocannabinoids, its use can be clearly differentiated with high-quality confirmatory testing—provided that appropriate analytical targets are included.

### 6.8. Differentiating Therapeutic Nabiximols Use from Recreational Cannabis Consumption

Distinguishing between prescribed medical use of nabiximols (a standardized oromucosal spray containing THC and CBD in an approximate 1:1 ratio) and recreational consumption of natural cannabis presents a complex challenge in clinical and forensic toxicology. While both sources originate from the cannabis plant, their phytochemical composition, administration routes, and pharmacokinetic profiles differ significantly, providing a basis for analytical differentiation.

Nabiximols is typically administered oromucosally in controlled doses, resulting in slower and more consistent absorption compared to inhaled cannabis. Peak plasma concentrations for both THC and CBD occur 30 min to 2 h post-administration, with lower interindividual variability than smoking or vaping. In contrast, recreational cannabis is often smoked or vaporized, leading to rapid THC absorption, higher initial peak concentrations, and a shorter duration of detection in blood [262].

The defining feature of nabiximols is its consistent 1:1 ratio of THC to CBD, which contrasts with the variable and often THC-dominant cannabinoid profiles found in most recreational cannabis products. Upon metabolism, both THC and CBD produce distinct metabolites detectable in biological matrices. The co-detection of CBD and its metabolites, when present at concentrations comparable to THC metabolites, may suggest pharmaceutical use of nabiximols, whereas low or undetectable CBD levels point toward recreational cannabis with minimal CBD content [278]. In addition, minor cannabinoids such as cannabigerol (CBG), cannabichromene (CBC), and tetrahydrocannabivarin (THCV) may be present in variable quantities in unregulated or illicit cannabis products, but are absent or present only in trace amounts in nabiximols formulations, providing further differentiation [276].

Advanced analytical methods, particularly LC–MS/MS, are widely used to quantify THC, CBD, their primary metabolites (e.g., 11-OH-THC, THC-COOH, 7-OH-CBD, 7-COOH-CBD), and minor cannabinoids with detection limits in the low ng/mL range [262,276]. Differentiation between nabiximols use and recreational cannabis is supported by cannabinoid ratios: therapeutic use typically yields CBD/THC ratios near 1, while recreational use shows THC dominance. Detection of minor plant cannabinoids or degradation products may further indicate non-pharmaceutical cannabis. Though techniques such as enantioselective analysis and isotope ratio mass spectrometry offer additional specificity, they are not yet standard in routine toxicology. While emerging methods such as enantiomeric profiling and isotope ratio mass spectrometry may offer additional discrimination power, they are not yet routinely implemented in most toxicology settings [262].

In legal and clinical contexts, analytical results must be interpreted in conjunction with case history, including medical prescriptions, dosing schedules, and timing of sample collection. The predictable cannabinoid ratio profile and limited presence of plant-derived impurities in nabiximols use provide strong objective markers for therapeutic administration. Conversely, variable cannabinoid content, high THC dominance, and co-detection of non-standard cannabinoids are suggestive of recreational use. Importantly, therapeutic use of nabiximols is typically documented and traceable, aiding forensic interpretation, whereas recreational use often lacks such contextual anchors. As cannabinoid-based medicines become more widely prescribed, precise analytical differentiation will remain essential for accurate exposure assessment, compliance monitoring, and legal adjudication.

In conclusion, distinguishing between the medical use of cannabinoids and their non-medical, recreational use remains a significant challenge in clinical and forensic toxicology. Despite advances in analytical techniques such as LC–MS/MS and highly selective chromatographic methods, the interpretation of results requires a thorough understanding of cannabinoid pharmacokinetics and metabolism. Reliable differentiation depends not only on biochemical profiles but also on the context of use, the presence of medical documentation, and the identification of specific metabolic markers. As cannabinoid-based medicines become more widely prescribed, the need for clear distinction between therapeutic use and misuse becomes increasingly important to ensure accurate diagnosis, protect patients, and support fair legal assessment.

## 7. Cocaine

Cocaine is a powerful central nervous system stimulant derived from the leaves of *Erythroxylum coca*, though synthetic analogues do not exist in the way opioids have. It exerts its primary effects by blocking monoamine transporters, especially the DAT) and to a lesser degree the SERT and NET transporters leading to elevated synaptic monoamine levels and consequent psychomotor stimulation, euphoria, and sympathomimetic effects. Cocaine also interacts with sigma (σ_1_ and σ_2_) receptors, which modulate dopaminergic neurotransmission and gene expression, including effects on dopamine D_1_/D_2_ receptor signaling and downstream pathways such as cAMP and ERK/MAPK cascades. These molecular interactions contribute to both its acute psychoactive properties and its neuroadaptations with repeated use [280,281].

Therapeutic uses of cocaine are extremely limited. Historically it was used as a local anesthetic (particularly in nasal or ophthalmologic procedures) and vasoconstrictor, but due to its high abuse potential and risk profile it has been largely replaced. The clinical relevance today is almost exclusively in understanding intoxication, toxicity, and managing dependence rather than direct therapeutic application.

In contemporary clinical practice cocaine is used almost exclusively as a topical agent for mucosal anesthesia and local vasoconstriction (e.g., otolaryngologic procedures of the nasal cavity and nasopharynx). In these applications cocaine is administered as a well-defined hydrochloride solution or impregnated pledget, in measured volumes and concentrations under clinician control; the objective is local (not systemic) anesthesia and bleeding control. Regulatory and clinical reviews emphasize this narrow, controlled indication and recommend specific handling and monitoring because of known systemic cardiovascular effects if absorbed [282].

Illicit use employs a broader set of routes chosen to modulate onset, intensity and duration of psychoactive effects. Common routes are: (1) intranasal insufflation (“snorting”) of the hydrochloride salt; (2) smoking of freebase/crack (rapid pulmonary absorption and very fast CNS exposure); (3) intravenous injection (immediate systemic bioavailability); and (4) oral ingestion (slower, more erratic absorption). Each route yields a characteristic time course of plasma concentration and subjective effect; smoking and injection produce the fastest rise in brain cocaine concentrations and the most intense, short-lived “high,” features strongly associated with higher abuse liability [283].

Prolonged or heavy cocaine use leads to a constellation of adverse effects. Cardiovascular complications are among the most serious: arrhythmias, myocardial ischemia, hypertension, strokes. Neurologic issues include seizures, cognitive impairment, risk of neurodegeneration in some studies. Psychiatric effects (paranoia, anxiety, depression, psychotic symptoms) are common. On the molecular level, repeated cocaine exposure produces tolerance (reduced response), sensitization (in some behavioral domains), dysregulation in dopamine receptor signaling (notably altered D_1_/D_2_ balance via sigma receptor modulation), and changes in gene expression that support craving and relapse [280,281,284].

In recognition of the above, the Diagnostic and Statistical Manual of Mental Disorders, Fifth Edition (DSM-5) defines Cocaine Use Disorder as a pattern of problematic use leading to clinically significant impairment or distress, manifesting at least 2 of 11 criteria within a 12-month period (including craving, tolerance, withdrawal, loss of control, neglect of roles, among others). Severity is graded as mild (2–3 symptoms), moderate (4–5), or severe (6 or more) [285].

Epidemiologically, cocaine use and related harms have been increasing. According to the 2023 World Drug Report, global cocaine production reached record levels, alongside rising prevalence. Approximately 25 million people used cocaine worldwide in 2023. The global burden of death and disability from Cocaine Use Disorder has risen dramatically over recent decades: deaths due to Cocaine Use Disorder increased several-fold from 1990 to 2019; age-standardized mortality and DALYs in high socio-demographic index countries have shown especially large increases [286,287]. In the United States, overdose deaths involving cocaine have increased greatly in recent years, both alone and often in combination with opioids; between 1999 and 2023, cocaine-involved overdose mortality rates rose from about 1.37 per 100,000 to 8.79 per 100,000 [288]. Also, a large proportion of cocaine-related overdose deaths involve opioid co-involvement [289].

Currently, there are no medications approved specifically for treating Cocaine Use Disorder. Several pharmacotherapeutic approaches have been evaluated. Disulfiram has shown some efficacy in increasing the number of people abstinent at end of treatment compared with placebo, but its effects on frequency or amount of use are mixed, and side effects are of concern [290]. Behavioral treatments including cognitive behavioural therapy, contingency management, intensive outpatient and group counseling remain the mainstay of treatment. Novel modalities are under study, for example transcranial magnetic stimulation has been trialled in randomized, double-blind sham-controlled designs for reducing craving and use, though results are as yet not definitive [291].

Despite the availability of these treatment options, access and effectiveness remain limited. Relapse rates are high as many individuals require multiple treatment episodes. Public health surveillance suggests under-reporting and under-treatment: for many countries, only small fractions of individuals with Cocaine Use Disorder receive evidence-based care. The overlapping epidemics of polysubstance use (especially with opioids and synthetic substances) compounds risk, including risk of overdose [287,289].

### 7.1. Cocaine Pharmacokinetics

The primary cocaine metabolites differ in their stability and detection windows: while cocaine’s plasma half-life is approximately 0.7–1.5 h, BE persists in urine for up to 2–3 days, and cocaethylene or norcocaine may be present even longer in chronic users [292,293,294]. Table 14 summarize the pharmacokinetic characteristics, analytical biomarkers, detection matrices and regulatory cut-off limits of cocaine [292,293,294,295,296,297,298,299,300,301,302,303]. It should be noted that cocaethylene detection indicate concomitant ethanol intake [304,305].

### 7.2. Toxicochemical Analysis and Metabolic Profiling of Cocaine

In contemporary biomedical and forensic practice, toxicochemical analysis of cocaine has evolved into an indispensable tool extending far beyond traditional forensic toxicology. Its scope now encompasses clinical diagnostics, workplace drug monitoring, law enforcement, military readiness assessment, and anti-doping control in sports medicine, where analytical results may carry significant clinical and legal consequences [283]. In clinical and emergency settings, cocaine testing is essential for confirming acute intoxication, guiding management of cardiovascular and neurological complications, and distinguishing therapeutic from illicit exposure. Moreover, in psychiatry and addiction medicine, objective confirmation of abstinence or relapse provides a foundation for treatment evaluation and compliance monitoring [303,306].

Routine laboratory detection of cocaine and its metabolites primarily rely on immunoassay screening, which detects either the parent compound or characteristic metabolites such as benzoylecgonine (BE) and ecgonine methyl ester (EME). Immunoassays are favored for their rapidity and low cost but remain limited by potential cross-reactivity with structurally related substances (e.g., cocaethylene or some local anesthetics) and by their inability to quantify specific analytes [307]. Therefore, confirmatory techniques, predominantly GC-MS or LC-MS/MS, are required to ensure analytical specificity and quantification accuracy [308,309]. These confirmatory platforms allow simultaneous detection of multiple cocaine metabolites—critical for distinguishing acute use from residual presence or environmental exposure.

Recent analytical innovations include high-resolution targeted cocaine metabolite panels capable of identifying more than twenty cocaine-related analytes in biological matrices such as urine, plasma, and hair [310]. Such panels routinely include cocaine, benzoylecgonine, ecgonine methyl ester, norcocaine, cocaethylene, and anhydroecgonine methyl ester (AEME), the latter serving as a biomarker of smoked “crack” cocaine exposure [311,312]. Advanced mass spectrometric workflows with isotopic internal standards and non-hydrolytic extraction have improved detection limits, reproducibility, and throughput, aligning with the increasing demands of clinical toxicology and forensic verification. Table 15 summarize the principal cocaine-related analytes.

The pharmacokinetic fate of cocaine is further complicated by pharmacogenetic variability in metabolic enzymes. Genetic polymorphisms in *BCHE* can markedly reduce catalytic efficiency, predisposing individuals to prolonged toxicity or increased cardiovascular risk [313].

Analytical interpretation must also account for matrix differences: while urine provides a wider detection window (up to 3 days in occasional users, longer in chronic users), blood and oral fluid are more suitable for determining recent use or impairment. Hair testing offers retrospective surveillance spanning weeks to months but requires rigorous contamination control [314].

### 7.3. Biomarkers, Analytical Thresholds, and Interpretive Challenges in Cocaine Testing

In conventional immunoassay-based drug screening BE, the principal urinary metabolite of cocaine serves as the primary target and calibrator. Beyond functioning as a direct marker of cocaine exposure, BE provides an indirect indicator of recent use since the parent compound is rapidly metabolized and typically present only in trace concentrations [315,316]. Most commercial assays therefore quantify results in “BE equivalents.” In clinical and workplace testing, a 300 ng/mL cutoff is commonly applied, although this may not reliably detect occasional or low-level users. Under federal workplace testing guidelines confirmatory thresholds of 100–150 ng/mL are adopted to minimize false positives [317].

The main limitation of BE as a single biomarker is its inability to differentiate active ingestion from passive or environmental exposure. Additionally, immunoassays can exhibit cross-reactivity with structurally similar compounds or metabolites such as cocaethylene or norcocaine, leading to ambiguous results [318]. Advanced immunoassays have improved specificity, but confirmatory chromatographic methods, particularly GC–MS and LC–MS/MS remain the gold standard for forensic confirmation due to their superior selectivity and quantification accuracy [309,319].

Because of their distinct formation mechanisms, certain metabolites serve as route-specific or co-ingestion biomarkers. The detection of AEME strongly supports crack cocaine smoking, while cocaethylene, a transesterification product with long half-life formed in the liver when cocaine is co-administered with ethanol, confirms concurrent ethanol use [320,321]. Thus, a multi-analyte approach that includes both primary and secondary metabolites significantly enhances interpretive reliability compared to reliance on BE alone.

Analytically, confirmatory testing commonly employs hydrolysis and derivatization steps to improve chromatographic resolution and detect glucuronide conjugates [319]. State-of-the-art methods such as UHPLC–MS/MS now enable simultaneous quantification of cocaine, BE, EME, AEME, and cocaethylene in biological matrices with limits of quantification in the low nanogram-per-milliliter range [310].

Finally, specimen type significantly influences interpretive conclusions. Urine remains the most widely used matrix due to its non-invasive collection and relatively long detection window. Blood and oral fluid better reflect recent intake and impairment but are temporally constrained [309]. Hair testing provides retrospective profiling over weeks to months but requires careful decontamination to eliminate environmental contamination [322].

## 8. Ketamine

Ketamine is a dissociative anesthetic belonging to the arylcyclohexylamine class, originally synthesized in 1962 as a safer alternative to phencyclidine. It acts as a noncompetitive antagonist of the N-methyl-D-aspartate (NMDA) receptor, thereby reducing excitatory glutamatergic neurotransmission in cortical and limbic circuits [323]. Beyond NMDA antagonism, ketamine interacts with a range of molecular targets including AMPA receptors, opioid receptors (μ and κ), monoamine transporters, muscarinic cholinergic receptors, and voltage-gated calcium channels, accounting for its diverse pharmacological profile [324]. Through these mechanisms, ketamine produces dose-dependent effects ranging from analgesia and dissociation to anesthesia and profound psychotomimetic experiences.

### 8.1. Therapeutic Uses and Mechanisms of Action

Clinically, ketamine has long been used as a short-acting dissociative anesthetic, especially in trauma, pediatric, and field medicine due to its preservation of airway reflexes and cardiovascular stability. In subanesthetic doses, ketamine has emerged as a rapid-acting antidepressant and analgesic, revolutionizing treatment paradigms for major depressive disorder, treatment-resistant depression, and chronic pain syndromes [325,326].

Mechanistically, the antidepressant effects of ketamine are attributed to transient NMDA receptor blockade on GABAergic interneurons, resulting in disinhibition of glutamate release, activation of AMPA receptors, and subsequent engagement of the BDNF-TrkB-mTOR signaling pathway [327]. These downstream cascades promote synaptogenesis and synaptic plasticity, providing a biological basis for ketamine’s rapid mood-restorative properties, effects not shared by conventional monoaminergic antidepressants.

Ketamine also exerts anti-inflammatory and neuroprotective effects through inhibition of microglial activation and reduction in proinflammatory cytokines (IL-6, TNF-α, IL-1β) [328]. These actions have fueled investigation into its potential roles in neurodegenerative and post-traumatic conditions, although long-term safety remains an active area of study.

### 8.2. Illicit Use and Misuse Patterns

Despite its therapeutic potential, ketamine is widely misused as a psychoactive recreational drug, commonly referred to as “Special K.” Illicit users typically seek out-of-body experiences, perceptual distortions, and feelings of detachment from self and environment, phenomena often described as “entering the K-hole” [329].

Illicit ketamine is typically diverted from medical sources or synthesized clandestinely, appearing as a white crystalline powder or liquid injectable formulation. Common routes of non-medical administration include intranasal insufflation (“snorting”), oral ingestion, intramuscular injection, and, less frequently, inhalation of vaporized powder [330].

In addition to recreational misuse, ketamine has gained notoriety as a chemical agent used in DFSA colloquially known as a “date-rape drug.” Its colorless, odorless, and slightly bitter-tasting liquid form can be easily mixed into beverages without detection. The rapid onset of action (within 5–10 min orally or intranasally) and its characteristic amnestic and sedative properties render victims disoriented, immobile, and often unable to recall events, thereby facilitating sexual assault [331].

Epidemiologically, non-medical ketamine use has increased globally, particularly in East and Southeast Asia, the United Kingdom, and North America. According to the UNODC World Drug Report 2023, seizures of illicit ketamine rose by over 250% between 2015 and 2022, with significant trafficking from East Asia to Europe and Oceania. Ketamine has also been identified as one of the most common novel psychoactive substances encountered in forensic toxicology in nightlife-related contexts [332].

### 8.3. Pharmacokinetics, Analytical Detection and Toxicochemical Profiling

Ketamine is a highly lipophilic compound that is rapidly absorbed and widely distributed throughout the body. Intravenous administration produces effects within 30–60 s, intramuscular within 2–5 min. It has relatively short duration of psychoactive effects that make it suitable for both anesthetic and rapid-acting antidepressant use. The most important pharmacokinetic properties are shown in Table 16 [324,333,334,335,336,337,338].

Laboratory identification of ketamine exposure typically involves immunoassay screening followed by GC-MS or LC-MS/MS confirmation. Immunoassays detect parent ketamine and norketamine but may show cross-reactivity with structurally related anesthetics [339]. LC-MS/MS methods allow simultaneous quantification of ketamine, norketamine, dehydronorketamine, and hydroxynorketamine in biological matrices such as plasma, urine, oral fluid, and hair [340]. A single positive ketamine test cannot be labeled as “medical” versus “illicit” without context but patterns of analytes, concentration ratios, matrix selection, and ancillary findings (co-drugs, excipients/impurities) provide strong forensic weight of evidence [324].

Clinical esketamine nasal spray is the S-enantiomer (pure S-ketamine). Detection of a strong S-enantiomer predominance (or near-pure S) in a sample strongly suggests use/diversion of pharmaceutical esketamine rather than typical illicit racemic ketamine (which is ~50:50 R:S). Illicit ketamine sold on the street is usually racemic (R and S present about equally), though some clandestine products vary depending on synthesis route [341].

High parent ketamine relative to norketamine in blood or oral fluid suggests very recent administration (especially IV or smoked routes), because parent peaks quickly after rapid systemic delivery. Lower parent—metabolite ratio (higher norketamine relative to parent) is consistent with oral dosing or delayed sampling (first-pass metabolism increases metabolite proportion). Intranasal tends to give intermediate parent/metabolite ratios (faster absorption than oral, but some first-pass still possible). These pharmacokinetic patterns have been reported in controlled administration studies and are useful for temporal/route inferences when combined with known PK half-lives [342,343].

Pharmaceutical formulations (e.g., commercially manufactured esketamine nasal spray or injectable ketamine) contain characteristic excipients and specific impurity profiles. Advanced forensic screens can detect formulation excipients/preservatives or the absence/presence of solvent residues consistent with clandestine synthesis.

Illicit powders may contain cutting agents (caffeine, mannitol, lidocaine, benzocaine, sugars) or synthesis impurities that are rarely present in legitimate medical products. Profiling such impurities (GC-MS, LC-MS) or performing non-targeted impurity profiling supports an illicit origin determination [344,345]. Co-detection of ethanol, benzodiazepines, GHB, or other club drugs may indicate recreational polysubstance use or drug-facilitated assault scenarios. This contextual information strengthens an inference of illicit/nonmedical use when combined with metabolic and enantiomeric data [324]. Table 17 shows the main pharmacokinetic characteristics, analytical biomarkers and detection matrices of ketamine.

### 8.4. Toxicology and Adverse Effects

Acute toxicity manifests as hypertension, tachycardia, nystagmus, ataxia, hallucinations, and occasionally respiratory depression at high doses. Unlike opioids, ketamine rarely causes fatal respiratory arrest in isolation. Chronic exposure, however, can produce urothelial damage due to excretion of irritant metabolites such as norketamine, resulting in ulcerative cystitis and fibrosis of the bladder wall [337].

At the neurochemical level, repeated ketamine exposure leads to dopaminergic dysregulation, oxidative stress, and neuroapoptotic signaling in animal models [338]. Cognitive sequelae include impairments in working memory and executive function, attributed to NMDA receptor hypofunction and prefrontal cortex dysconnectivity [329].

## 9. Application of Metabolomics in Differentiating Legal Therapeutic Use from Illicit Drug Abuse

Metabolic profiling provides valuable interpretive insight for distinguishing therapeutic compliance from misuse or illicit drug intake. Parent-to-metabolite ratios facilitate the assessment of dosing consistency and temporal patterns of use, while chiral analysis enhances diagnostic specificity by differentiating between enantiomeric formulations associated with prescribed medications and those resulting from illicit synthesis, as demonstrated for L- and D-methamphetamine or in the enantiomeric distribution of zopiclone metabolites [83]. Moreover, secondary markers such as unique glucuronide conjugates or synthesis by-products can serve as indicators of legitimate metabolism versus counterfeit formulations, thereby supporting both clinical monitoring and forensic interpretation [346]. Beyond traditional analytical methods, modern toxicochemistry increasingly employs innovative mass spectrometric approaches, including ambient ionization and high-resolution ion mobility spectrometry, enabling rapid, minimally prepared, and structurally resolved analyses. Mass spectrometry imaging further provides spatial distribution data for narcotic compounds within biological tissues, facilitating a multidimensional understanding of drug disposition [347,348]. Complementary to these are NMR-based metabolomics and surface-enhanced Raman scattering, which support structural elucidation of novel psychoactive substances [349]. Advances in computational mass spectrometry, including in silico fragmentation prediction, molecular networking, and machine learning–assisted identification, have significantly improved the annotation of unknown metabolites and novel psychoactive substances [350]. Although these emerging methods do not yet replace chromatographic gold standards, they substantially expand analytical capacity for rapid, sensitive, and multidimensional assessment of drug misuse in clinical and forensic contexts. Additionally, the emergence of new psychoactive substances, notably designer benzodiazepines and synthetic cathinones poses significant analytical challenges due to rapid structural evolution, limited toxicological data, and absence of reference standards. Their complex metabolic pathways generate multiple low-abundance metabolites, often below detection thresholds of conventional GC–MS or LC–MS assays. HRMS and non-targeted metabolomic workflows now enhance detection and structural elucidation of these compounds, supporting more accurate differentiation between legitimate pharmaceutical analogues and illicit designer derivatives [351].

Despite their diagnostic utility, drug metabolites are often rapidly eliminated, complicating detection. Conversely, alterations in endogenous metabolites tend to persist longer, offering indirect yet robust markers of exposure. In this regard, metabolomics has emerged as a powerful tool for differentiating between therapeutic drug use and drug abuse [352]. While classical toxicochemical analysis relies on detecting xenobiotics and their metabolites, metabolomics captures systemic biochemical perturbations induced by drug exposure. These endogenous signatures reflect dose- and route-dependent metabolic adaptations and can differentiate chronic abuse from legitimate use [353]. For instance, supervised opioid therapy induces transient changes in energy metabolism, whereas opioid abuse leads to persistent disruptions in lipid and oxidative pathways [354]. Comparable metabolic distinctions have been described for stimulants, benzodiazepines, and cannabinoids [355,356].

## 10. Conclusions

Metabolite profiling provides a scientifically robust means of distinguishing therapeutic from illicit drug use by examining both parent compounds and their biotransformation products across biological matrices. This approach enables the differentiation between legitimate medical use, misuse, and illicit consumption with a degree of specificity unattainable by conventional screening methods. Moreover, as analytical technologies continue to evolve, metabolite-based differentiation has the potential to become a cornerstone of modern clinical and forensic toxicology. Beyond its diagnostic precision, metabolite profiling also supports public health efforts by improving drug monitoring programs and may be useful in guiding policy decisions aimed at minimizing substance misuse while ensuring appropriate therapeutic access.

Despite the significant potential of metabolite profiling to distinguish therapeutic drug use from abuse, this approach has several key limitations. First, the required instrumentation and sample preparation procedures are costly and require specialized technical expertise, which limits broad clinical or forensic implementation. Second, analytical methods cannot capture the entire metabolome: metabolites exhibit a wide range of concentrations and chemical properties, and no single platform can simultaneously detect all metabolites (from polar to non-polar, from low to high abundance) with equal sensitivity and specificity. Third, metabolomic data are high-dimensional and complex, requiring advanced bioinformatic approaches and expert knowledge, with potential risks of errors or misinterpretation. Therefore, when interpreting metabolomic profiles for distinguishing therapeutic use from drug abuse, these limitations must be considered, including the possibility of false positives or negatives and the need for appropriate controls and validation.

## Figures and Tables

**Figure 1 metabolites-15-00745-f001:**
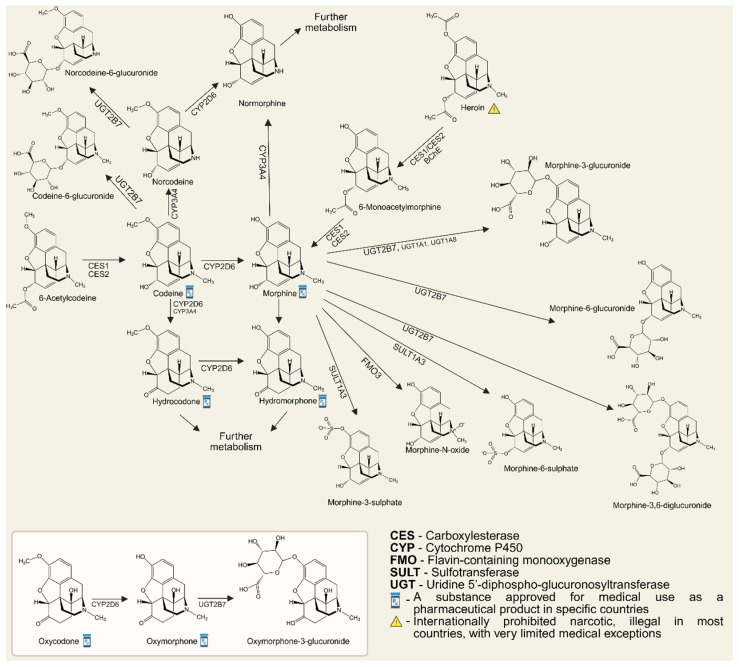
Simplified metabolic pathways of major opioids and their shared metabolites.

**Figure 2 metabolites-15-00745-f002:**
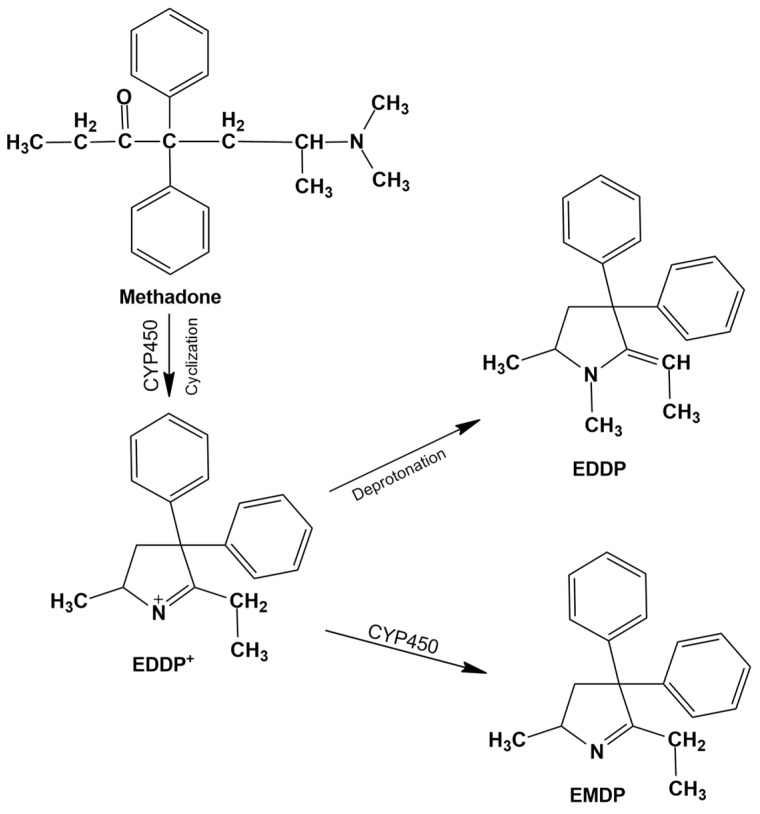
Methadone metabolism and main metabolites.

**Figure 3 metabolites-15-00745-f003:**
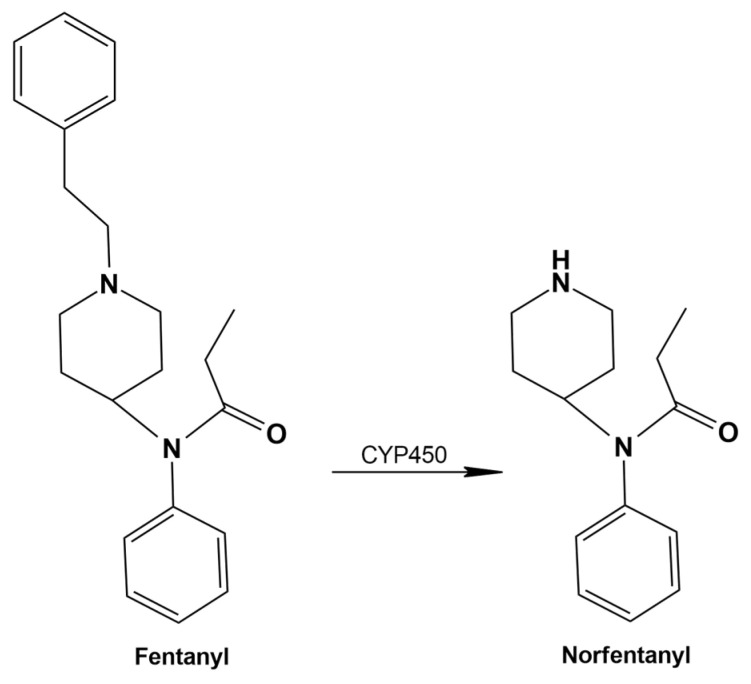
Main metabolic pathway of fentanyl leading to norfentanyl.

**Figure 4 metabolites-15-00745-f004:**
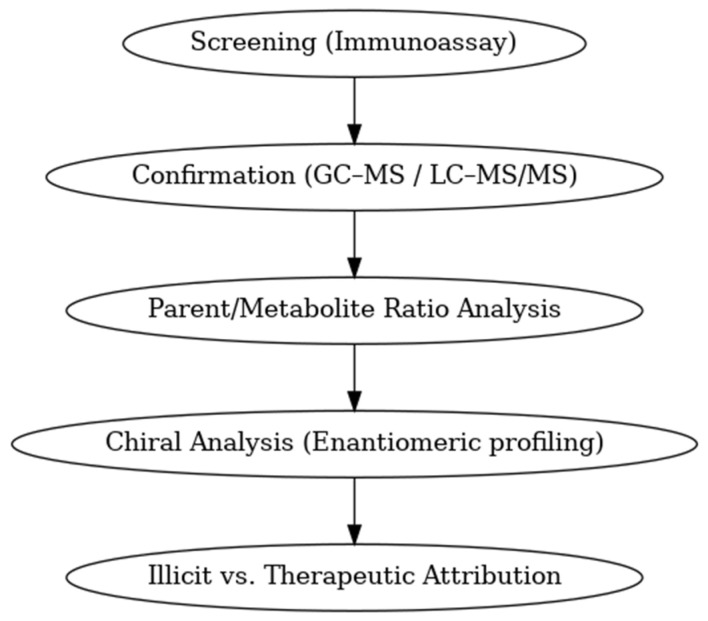
Simplified analytical workflow for distinguishing therapeutic from illicit ATS use.

**Figure 5 metabolites-15-00745-f005:**
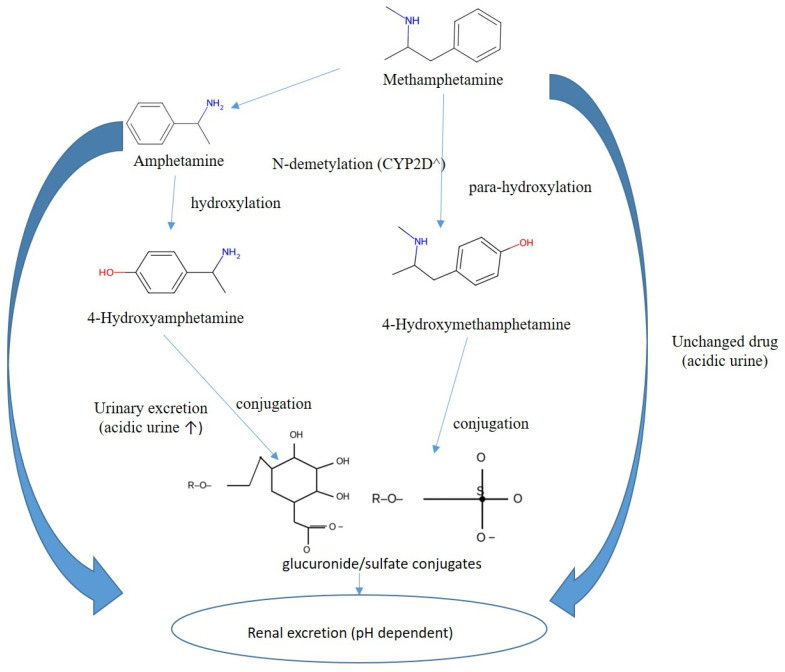
Overview of methamphetamine metabolism to amphetamine and hydroxy-metabolites with subsequent conjugation.

**Figure 6 metabolites-15-00745-f006:**
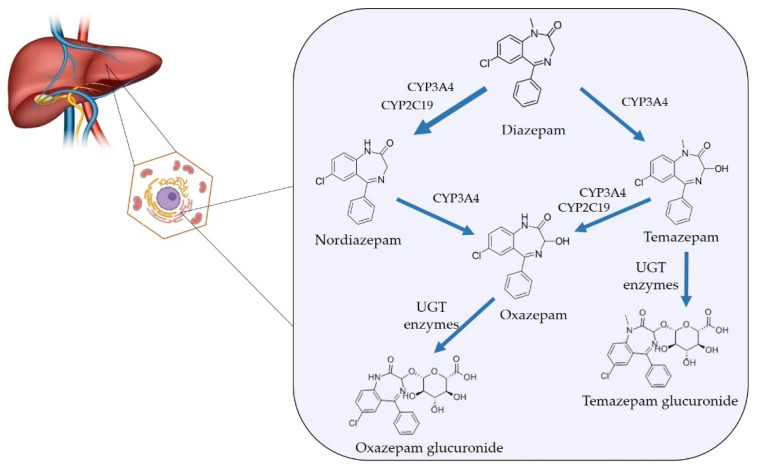
Diazepam metabolism in humans.

**Table 1 metabolites-15-00745-t001:** List of the thirty-three target analytes in the TOSU assay, including parent compounds and their metabolites [37,40,41].

Primary Molecule	Metabolite
**Codeine**	Codeine-6-β-glucuronide
**Morphine**	Morphine-6-β-glucuronide
**-**	6-Monoacetylmorphine (6-MAM) ^†^
**Hydrocodone**	Norhydrocodone
**Dihydrocodeine**	–
**Hydromorphone**	Hydromorphone-3-β-glucuronide
**Oxycodone**	Noroxycodone
**Oxymorphone**	Oxymorphone-3-β-glucuronide, Noroxymorphone
**Meperidine**	Normeperidine
**Methadone**	2-ethylidene-1,5-dimethyl-3,3-diphenylpyrrolidine (EDDP)
**Propoxyphene**	Norpropoxyphene
**Tramadol**	O-Desmethyltramadol
**Tapentadol**	Tapentadol-β-glucuronide, N-Desmethyltapentadol
**Buprenorphine**	Norbuprenorphine, Norbuprenorphine glucuronide
**Naloxone**	Naloxone glucuronide
**Fentanyl**	Norfentanyl

^†^ 6-MAM—A specific metabolite widely recognized as a biomarker of heroin use.

**Table 2 metabolites-15-00745-t002:** Pharmacokinetic characteristics, analytical biomarkers, detection matrices, regulatory cut-off limits, and major analytical challenges of selected opioids.

Opioid	Pharmacokinetics	Biomarkers of Use/Recommended Matrices and Analytical Methods	Urine Cut-Off Limits (Screening/Confirmatory, ng/mL)	Major Analytical Challenges
**Morphine**	Oral bioavailability ~20–40% Elimination half-life (t½): ~2–4 h (parent compound) Undergoes extensive hepatic glucuronidation via UGT2B7 to morphine-3-glucuronide (M3G) and morphine-6-glucuronide (M6G) Excreted renally as parent and conjugated metabolites	Biomarkers: morphine (parent), M3G, M6G Matrices: urine (detectable for several days), blood/plasma (reflects impairment, hours), oral fluid (recent exposure), hair (long-term monitoring) Methods: immunoassay screening followed by LC-MS/MS or GC-MS confirmation (after enzymatic hydrolysis when quantifying glucuronides)	Initial: 2000 ng/mL (group opiate screen) Confirmatory: 4000 ng/mL (morphine)	Cross-reactivity in immunoassays (other opioids, glucuronides) Requirement for deconjugation/hydrolysis prior to quantification Matrix effects in hair analysis Differentiation between M6G and M3G (active vs. inactive) Post-collection hydrolysis and stability issues
**Codeine**	Oral bioavailability ~60–80% Elimination half-life ~3–4 h Metabolized via O- and N-demethylation: CYP2D6-mediated conversion to morphine (polymorphic), and formation of norcodeine and codeine-6-glucuronide (C6G)	Biomarkers: codeine (parent), morphine (metabolite indicating O-demethylation), norcodeine, C6G Matrices: urine, blood, oral fluid, hair Methods: immunoassay screening (opioid panel) followed by confirmatory LC-MS/MS; chiral analysis if source discrimination is required	Initial: 2000 ng/mL (group opiate screen) Confirmatory: 2000 ng/mL (codeine)	Differentiation between codeine use and morphine/heroin exposure CYP2D6 polymorphism causes large interindividual metabolic variability Potential cross-reactivity in immunoassays
**Heroin**	Extremely rapid hydrolysis: heroin → 6-MAM → morphine Plasma t½: minutes 6-MAM detectable for only minutes to hours; morphine persists longer	Biomarkers: 6-MAM (pathognomonic for heroin), morphine, and its glucuronides Matrices: blood/plasma (very short detection window for 6-MAM), urine (6-MAM detectable up to ~24 h post-use), hair (long-term incorporation) Methods: LC-MS/MS or GC-MS with rapid sample handling; high-resolution MS for low-level 6-MAM	Initial/Confirmatory: 10 ng/mL (6-MAM)	Pronounced chemical and enzymatic instability (ex vivo hydrolysis to morphine) Necessity for immediate sample preservation (cooling, enzyme inhibitors) High risk of false negatives for 6-MAM if sample handling delayed Interpretation complicated when only morphine is present (shared metabolic origin with morphine and codeine)
**Oxycodone/Oxymorphone/Hydrocodone/Hydromorphone**	Oxycodone: t½ ~3–4 h Oxycodone metabolized by CYP3A4 → noroxycodone and CYP2D6 → oxymorphone (active) Hydrocodone metabolized to hydromorphone (CYP2D6) Hydromorphone has shorter t½ • Extensive hepatic metabolism and renal excretion	Biomarkers: parent drugs (oxycodone, hydrocodone, hydromorphone, oxymorphone) and metabolites (noroxycodone, norhydrocodone) Matrices: urine, blood, oral fluid, hair Methods: LC-MS/MS required due to poor immunoassay sensitivity and selectivity; chiral or high-resolution MS used to resolve structural analogues	Oxycodone/Oxymorphon: Initial: 100 ng/mL; Confirmatory: 100 ng/mL Hydrocodone/Hydromorphone: Initial: 300 ng/mL; Confirmatory: 100 ng/mL	Cross-reactivity and variable immunoassay sensitivity across semisynthetic opioids Need for targeted MS/MS transitions for structural analogues Parent/metabolite ratios affected by CYP2D6 polymorphism and drug interactions (inducers/inhibitors)
**Methadone**	High but variable oral bioavailability Very long and variable elimination half-life (8–59 h; enantiomer-dependent) Metabolized hepatically (CYP3A4, CYP2B6, others) → EDDP, EMDP Excreted renally and biliary	Biomarkers: methadone (parent) and EDDP (primary metabolite indicating use/adherence) Matrices: blood/plasma (therapeutic monitoring), urine (compliance/toxicity), hair (chronic use) Methods: immunoassay screening (methadone-specific) followed by confirmatory LC-MS/MS; chiral separation if stereochemical profiling required	Initial: 300 ng/mL (typical immunoassay screening) Confirmatory: 300 ng/mL (methadone and/or EDDP; program-dependent). Some EDDP assays use 100 ng/mL as qualitative threshold	Therapeutic and toxic plasma levels may overlap Long half-life complicates interpretation (accumulation) Co-medications affecting CYP3A4/2B6 alter EDDP formation Occasional need for chiral discrimination (forensic contexts) Variable incorporation and binding in keratin matrices (hair)

→ the symbol shows the metabolic pathway.

**Table 3 metabolites-15-00745-t003:** Cut-off values, common cross-reactants, and examples of false positives in ATS immunoassays.

Context/Program	Initial Screen Cut-Off (ng/mL)	Confirmatory Cut-Off (ng/mL)	Analytes Covered by Screen	Common Cross-Reactants/Interferents	Differentiation
Clinical/workplace (typical)	500–1000 (as amphetamine equivalents)	By LC-MS/MS, laboratory-defined (commonly 150–250)	Amphetamine, methamphetamine (variable cross-reactivity to MDMA/MDEA)	Pseudoephedrine/ephedrine, bupropion, atomoxetine, phentermine, metoprolol, labetalol metabolite (1-methyl-3-phenylpropylamine), ofloxacin, moxifloxacin	False positives most frequent with bupropion and labetalol metabolite; false negatives possible for ring-substituted ATS (e.g., MDMA) [134,137].
US Federal programs (SAMHSA)	500 (amphetamine class)	250 (amphetamine class)	Targets amphetamine class; MDMA/MDA often on separate panels	Pseudoephedrine/ephedrine; phenethylamines; DMAA (reported); selegiline → L-methamphetamine	Lower confirmatory cut-off reduces spurious positives; enantiomeric analysis helps distinguish L-methamphetamine from therapeutic selegiline vs. illicit D-methamphetamine [132,139].
Pediatric/adolescent screens	Assay-dependent (commonly 500)	Per laboratory	Amphetamine class	Aripiprazole; trazodone; promethazine; chlorpromazine (assay-dependent)	False positives with aripiprazole in youth; confirmation by GC-MS or LC-MS/MS is mandatory [136].
MDMA/MDEA (ring-substituted ATS)	Often poor cross-reactivity on amphetamine-calibrated screens	Targeted LC-MS/MS confirmation	MDMA/MDA/MDEA (on dedicated assays)	—	Generic amphetamine immunoassays may miss MDMA/MDEA; targeted panels required for detection [46].

Abbreviations: ATS, amphetamine-type stimulants; LC-MS/MS, liquid chromatography–tandem mass spectrometry; DOT, Department of Transportation.

**Table 4 metabolites-15-00745-t004:** Clinically and illicitly used amphetamine-type stimulants: main representatives.

Drug	Pharmacokinetic Properties	Major Metabolic Pathways and Key Metabolites	Notes
**Amphetamine**	Rapid oral absorption; T_max_ ≈ 3 h; t_1_/_2_ ≈ 9–14 h; renal clearance strongly pH-dependent (acidic urine ↑ excretion)	Oxidative deamination and p-hydroxylation mainly via CYP2D6; conjugation with sulfate and glucuronic acid	D-isomer is more potent on CNS than L-isomer; urinary acidification increases clearance; detection of high AM/MA ratio suggests primary amphetamine use
**Methamphetamine**	Rapid absorption (oral, nasal, intravenous); T_max_ ≈ 2–3 h; t_1_/_2_ ≈ 10–12 h (influenced by urinary pH)	N-demethylation (CYP2D6) → amphetamine; para-hydroxylation → 4-hydroxymethamphetamine and 4-hydroxyamphetamine → conjugation	MA/AM ratio > 2–3 indicates methamphetamine intake; alkaline urine prolongs elimination; chiral analysis distinguishes therapeutic L-methamphetamine (selegiline/decongestants) from illicit D-isomer
**Lisdexamfetamine**	Prodrug; enzymatic hydrolysis in blood; T_max_ ≈ 1 h for parent, 3.5 h for released d-amphetamine; t_1_/_2_ ≈ 8–13 h (as d-amphetamine)	Enzymatic hydrolysis to d-amphetamine and L-lysine; non-CYP-mediated metabolism	Lower abuse potential; absence of L-isomer confirms therapeutic origin; metabolite profile identical to d-amphetamine
**MDMA (3,4-Methylenedioxymethamphetamine)**	Oral T_max_ ≈ 2 h; t_1_/_2_ ≈ 6–10 h; extensive first-pass metabolism	O-demethylenation (CYP2D6) → MDA; O-methylation → HMMA, HMA; subsequent conjugation	MDMA/MDA ratio > 1 = recent use; CYP2D6 polymorphism or SSRI co-therapy delays MDA formation
**MDA (3,4-Methylenedioxyamphetamine)**	Oral T_max_ ≈ 1.5–2 h; t_1_/_2_ ≈ 8–12 h	O-demethylenation, hydroxylation, and conjugation → HMA, HMAA	Active metabolite of MDMA; predominance of MDA without MDMA indicates direct use
**MDEA (3,4-Methylenedioxyethylamphetamine)**	Oral T_max_ ≈ 1.5–2 h; t_1_/_2_ ≈ 6–9 h	O-demethylenation → DHEA; O-methylation → HMEA; minor N-deethylation → MDA	Detection of HMEA confirms MDEA intake; concurrent presence of MDEA and MDMA indicates polydrug use
**Mephedrone (4-MMC, 4-methylmethcathinone)**	Rapid absorption (oral/nasal); T_max_ ≈ 1–2 h; t_1_/_2_ ≈ 6–8 h	N-demethylation → nor-mephedrone; reduction → dihydromephedrone; para-hydroxylation	Parent compound unstable; detection indicates illicit use; no approved therapeutic application; detection = illicit exposure
**Methylone (bk-MDMA)**	Oral T_max_ ≈ 1–2 h; t_1_/_2_ ≈ 6–9 h	N-demethylation → 3,4-methylenedioxycathinone; O-methylation → HMMC	Overlaps with MDMA metabolites; requires HRMS for differentiation
**Methcathinone (ephedrone)**	Oral T_max_ ≈ 1–2 h; t_1_/_2_ ≈ 4–6 h	N-demethylation, reduction, and hydroxylation; conjugation to glucuronides	Structurally related to cathinone; rapid metabolism limits detection; illicitly synthesized from ephedrine
**Cathinone (natural from Catha edulis)**	Rapid absorption; short t_1_/_2_ ≈ 1.5 h	Reduction → cathine (norpseudoephedrine) and norephedrine	Naturally occurring stimulant; detection of cathine/norephedrine profile distinguishes natural vs. synthetic cathinones

↑—increase.

**Table 5 metabolites-15-00745-t005:** Parent/metabolite ratios (methamphetamine/amphetamine), influence of urine pH, forensic notes *.

Analyte(s)	Typical Parent/Metabolite Ratio	Influence of Urine pH	Forensic Notes
Methamphetamine (MA); Amphetamine (AM)	MA:AM usually >2–3:1 after methamphetamine intake	Acidic urine → faster excretion, lower ratio; Alkaline urine → slower excretion, higher ratio	Detection of AM may result from MA metabolism; High MA with some AM = methamphetamine use; High AM with little/no MA = primary amphetamine use [27,37].
Amphetamine (AM) (parent drug)	AM >> MA (if any MA detected at all)	Urine pH affects excretion rate of AM (faster in acidic, slower in alkaline urine), but no conversion to MA	Predominant AM with minimal/no MA indicates direct amphetamine intake [46].
Hydroxylated metabolites (e.g., 4-hydroxyamphetamine, 4-hydroxymethamphetamine)	Generally low compared to parent drug(s)	Also pH-dependent excretion	Supportive markers of metabolism but less critical for distinguishing AM vs. MA intake [140].
Chiral analysis	-	-	Consider chiral analysis when L-methamphetamine exposure (e.g., from selegiline therapy or decongestants) is plausible [143].

* Urinary acidification accelerates renal elimination of unchanged amphetamine and methamphetamine, whereas alkalinization slows elimination and may raise MA/AM ratios.

**Table 6 metabolites-15-00745-t006:** Parent drug MDMA and related metabolites: interpretive value for acute vs. late use and polydrug exposure.

Compound	Main Origin/Metabolism	Interpretive Significance
MDMA (3,4-Methylenedioxymethamphetamine, “Ecstasy”)	Parent drug; metabolized to MDA, HMMA	High MDMA with little metabolite → acute/recent use [42]
MDA (3,4-Methylenedioxyamphetamine)	Active metabolite of MDMA; also a parent drug	Presence with MDMA → expected metabolite. Predominant MDA may suggest direct MDA ingestion [43]
HMMA (4-Hydroxy-3-methoxymethamphetamine)	Major MDMA metabolite (O-demethylenation + O-methylation)	Typically higher in urine at later time points → late/ongoing excretion; supportive for MDMA intake [44]
MDEA (3,4-Methylenedioxyethylamphetamine, “Eve”)	Structurally related drug; metabolized to MDA	Detection indicates separate ingestion (polydrug or substitution) [45]
DHEA (3,4-Dihydroxyethylamphetamine)	Hydroxylated metabolite of MDEA	Confirms MDEA metabolism; interpret cautiously due to instability [46]
HMEA (4-Hydroxy-3-methoxyethylamphetamine)	O-methylated metabolite of MDEA	Supportive marker of MDEA use; usually detected later in urine [46]

**Table 7 metabolites-15-00745-t007:** Representative synthetic and natural cathinones: major metabolites and analytical challenges.

Parent Cathinone	Major Metabolites	Analytical Challenges	Differentiation
Mephedrone (4-MMC, 4-methylmethcathinone)	Nor-mephedrone (N-demethylated), 4-hydroxytolylmephedrone, dihydromephedrone	Structural similarity to other cathinones; extensive metabolism reduces parent detectability; instability in stored samples.	No therapeutic use; detection implies illicit exposure. Risk of misclassification with other cathinones without targeted LC-MS/MS [140,150].
Methylone (bk-MDMA, 3,4-methylenedioxy-N-methylcathinone)	3,4-methylenedioxycathinone (N-demethylated), HMMC (4-hydroxy-3-methoxymethcathinone)	Overlap with MDMA metabolites complicates interpretation; rapid metabolism; need for high-resolution MS.	Misattribution as MDMA/MDEA intake possible if only metabolite profile is assessed. Requires HRMS or targeted transitions [140,151].
Ethylone (bk-MDEA, 3,4-methylenedioxy-N-ethylcathinone)	MDEC (3,4-methylenedioxycathinone, N-deethylated), HMEEC (hydroxy-methoxyethylcathinone)	Difficult differentiation from methylone and MDEA use; low stability in biological matrices.	No approved therapeutic analogue; may be misinterpreted as MDEA ingestion without confirmatory HRMS [140,151].
α-PVP (alpha-pyrrolidinovalerophenone, “Flakka”)	Hydroxylated α-PVP, keto-reduced metabolites	Strong structural overlap with other pyrrolidinophenones; poor chromatographic separation; need for LC-HRMS/MS confirmation.	Lacks therapeutic indication; detection is highly specific for illicit/designer use [152,153].
Cathinone (natural, from khat)	Cathine (norpseudoephedrine), norephedrine	Parent unstable; low concentrations in biological specimens; co-use with synthetic cathinones complicates interpretation.	Only naturally occurring cathinone with traditional use; differentiation from synthetic analogues requires precise metabolite profiling [154,155].

**Table 8 metabolites-15-00745-t008:** Enantiomeric clues for ATS source attribution.

Finding	Interpretive Significance
Predominance of D-methamphetamine	Indicative of illicit or prescription D-methamphetamine use [139].
Predominance of L-methamphetamine	Supports exposure from selegiline metabolism or intranasal decongestants [46,127].
Detection of D-amphetamine only (e.g., from lisdexamfetamine)	Consistent with therapeutic adherence [46,127].
Enantiomeric profiles interpreted with MA/AM ratio, urinary pH, and time since intake	Provides contextual confirmation; prevents misattribution [130].

**Table 9 metabolites-15-00745-t009:** The main clinically used benzodiazepines and Z-drugs [171,172,173,174,175,176,177,178,179,180,181,182,183,184,185,186,187,188,189,190,191,192,193,194,195,196,197,198].

Drug	Pharmacokinetic Properties	Major Metabolic Pathways and Key Metabolites
**BENZODIAZEPINES**		
**Diazepam**	T_max_ ≈ 1–2 h Long T_1_/_2_ for parent drug ≈ 30–60 h;	CYP-mediated (CYP3A4, CYP2C19) N-demethylation to desmethyldiazepam (nordiazepam) (active) → oxazepam (active) → glucuronides; CYP-mediated N-hydroxylation to temazepam (active) → oxazepam → glucuronides
**Alprazolam**	T_max_ ≈ 1–2 h T_1_/_2_ ≈ 9–16 h (prolonged in elderly and in hepatic impairment)	CYP3A4 oxidation to α-hydroxyalprazolam (active, low levels), 4-hydroxyalprazolam and minor inactive metabolites
**Bromazepam**	T_max_ ≈ 1–3 h (oral) T_1_/_2_ ≈ 10–20 h (variable; prolonged in elderly)	Oxidation to 3-hydroxybromazepam and other metabolites and subsequent conjugation; urinary detection of bromazepam and its major metabolite is conducted after hydrolysis of the glucuronide conjugates
**Midazolam**	T_max_ ≈ 0.5–1.5 h (oral) T_1_/_2_ ≈ 1.5–6 h (single dose; prolonged in ICU/renal/hepatic impairment)	CYP3A4 hydroxylation to 1-hydroxymidazolam (major metabolite) and 4-hydroxymidazolam and subsequent biotransformation to glucuronides that are often measured in the urine
**Triazolam**	T_max_ ≈ 0.5–2 h T_1_/_2_ ≈ 1.5–5.5 h	Hydroxylation by CYP3A to α-hydroxytriazolam (active) and 4-hydroxytriazolam and conjugation to inactive metabolites
**Cinolazepam**	T_max_ ≈ 1–2 h (oral); variable T_1_/_2_ (limited data)	Metabolized to N-(hydroxyethyl) cinolazepam and glucuronides
**Lorazepam**	T_max_ ≈ 1–2 h T_1_/_2_ ≈ 10–20 h (variable)	Direct glucuronidation to lorazepam-glucuronide (inactive).
**Clonazepam**	T_max_ ≈ 1–4 h (varies with formulation); T_1_/_2_ ≈ 30–60 h (wide intersubject variability)	Nitro reduction to 7-aminoclonazepam and subsequent N-acetylation to 7-acetamidoclonazepam; minor hydroxylation products
**Nitrazepam**	T_max_ ≈ 1–2 h; good oral absorption T_1_/_2_ ≈ 16–48 h	Nitro reduction to 7-aminonitrazepam and subsequent N-acetylation to 7-acetamidonitrazepam (major urinary metabolites); no clinically significant active metabolites
**Clobazam**	T_max_ ≈ 1–2 h T_1_/_2_ ≈ 36–42 h (parent drug); T_1_/_2_ ≈ 59–74 h (active metabolite)	Oxidative N-demethylation by CYP450 enzymes (CYP3A4, CYP2C19) to N-desmethylclobazam (active); hydroxylation and glucuronidation
**Flunitrazepam**	T_max_ ≈ 1–2 h T_1_/_2_ ≈ 18–26 h (parent drug); Active metabolite is detectable in urine for days even weeks	Nitro-reduction to 7-aminoflunitrazepam and related desmethyl metabolites; conjugation → urinary metabolites
**Temazepam**	T_max_ ≈ 1–2 h T_1_/_2_ ≈ 8–20 h	CYP3A4 hydroxylation to oxazepam (active); glucuronidation
**Z-DRUGS**		
**Zolpidem**	T_max_ ≈ 1–2 h; T_1_/_2_ ≈ 1.5–3 h (↑ in elderly and in hepatic impairment)	Hydroxylation mainly via CYP3A4 (also CYP2C9, CYP1A2) to inactive metabolites; <1% is excreted unchanged in the urine
**Zaleplon**	T_max_ ≈ 0.7–1.4 h; T_1_/_2_ ≈ ~1 h (ultra-short)	Oxidation by aldehyde oxidase to 5-oxo-zaleplon (inactive) that is further dealkylated or N-dealkylation by CYP3A4 to N-desethyl-zaleplon that is further oxidated; glucuronidation
**Zopiclone/Eszopiclone**	T_max_ ≈ 1–2 h; T_1_/_2_ ≈ 5–7 h (longer than zolpidem)	N-demethylation to N-desmethylzopiclone (inactive) and zopiclone N-oxide (active); eszopiclone is oxidized and demethylated by CYP3A4 and CYP2E1 to inactive metabolites

**Table 10 metabolites-15-00745-t010:** Analytical biomarkers, detection matrices and methods and regulatory cut-off limits for selected benzodiazepines and Z-drugs.

Drug; Primary Biomarker(s)/Key Metabolites	Recommended Matrix	Detection Method/Detection Window	Typical Confirmatory Cut-Off or LOQ (Reported Range)	Notes
**BENZODIAZEPINES**				
**Diazepam** Biomarkers: Diazepam; nordiazepam; oxazepam; temazepam	Plasma/serum; urine; hair	LC–MS/MS (often with GC–MS backup); Detection window Plasma: hours to days Urine: days to weeks	~5–50 ng/mL (urine) LOQ/confirmatory cut-off varies across methods	Long-acting active metabolites prolong the detection window; metabolite/parent ratios are used to help estimate time since dosing
**Alprazolam** **Biomarkers:** Alprazolam; α-hydroxyalprazolam; 4-hydroxyalprazolam	Plasma; DBS; urine	LC–MS/MS (including DBS); Detection window Plasma: hours Urine: ~1–3 days	~0.05–5 ng/mL (LOQ (DBS/plasma); higher LOQ in urine)	Parent drug is largely responsible for the effect; CYP3A4 inhibitors markedly increase exposure so metabolite ratios are typically low
**Bromazepam** **Biomarkers:** Bromazepam; 3-hydroxybromazepam; minor desmethyl forms	Urine; plasma; hair	LC–MS/MS Detection window Urine: several days for 3-OH metabolite and conjugates	~5–20 ng/mL (urine)	3-hydroxybromazepam and conjugates are used as urinary markers; it is reported for them to be detectable for days depending on dose and method
**Midazolam** **Biomarkers:** Midazolam; 1′-hydroxymidazolam (active) and glucuronide	Plasma; urine	LC–MS/MS HPLC-UV Detection window Plasma: minutes to hours Urine: ~1–2 days	~1–10 ng/mL (urine)	1′-hydroxymidazolam and its glucuronide are the major analytes in plasma and urine; their concentrations and metabolite/parent ratios are strongly influenced by the duration of administration and organ function
**Lorazepam** **Biomarkers:** Lorazepam; lorazepam-3-O-glucuronide	Urine (after hydrolysis); plasma; hair	LC–MS/MS (in urine after hydrolysis) Detection window Plasma: hours to days Urine: days	~5–50 ng/mL (urine; glucuronide measurement may require hydrolysis)	Lack of active metabolites makes lorazepam less likely to accumulate; lorazepam and its glucuronide metabolites are detected in urine
**Clonazepam** **Biomarkers:** Clonazepam; 7-amnoclonazepam	Plasma; urine; hair	LC–MS/MS Detection window Urine: days to weeks for metabolite	~1–20 ng/mL (urine)	7-aminoclonazepam is a major urinary metabolite used in forensic/toxicology screens
**Nitrazepam** **Biomarkers:** Nitrazepam; 7-aminonitrazepam	Urine; hair; plasma	LC–MS/MS Detection window Urine: days to weeks for 7-aminonitrazepam	~1–20 ng/mL (urine)	7-aminonitrazepam is the primary forensic marker; acetylated derivatives are commonly detected after hydrolysis
**Flunitrazepam** **Biomarkers:** Flunitrazepam; 7-aminoflunitrazepam	Urine; hair; plasma	LC–MS/MS Detection window Urine: days to weeks for 7-aminoflunitrazepam	~1–20 ng/mL (urine LOQ); hair: pg/mg—low ng/mg depending on the method	7-aminoflunitrazepam is the key forensic marker (often detectable long after parent drug); used in “date-rape” forensic investigations—metabolite ratios may help estimate time since ingestion
**Z-DRUGS**				
**Zolpidem** **Biomarkers:** Zolpidem; oxidative metabolites (phenyl-4-carboxylic acid)	Plasma; urine; hair	LC–MS/MS and immunoassay screening methods Detection window Plasma: hours Urine: ~1–2 days (metabolites)	~0.5–10 ng/mL (urine)	Zolpidem has short detection window; metabolites are generally inactive; parent drug concentration reflects recent intake (useful to differentiate recent therapeutic dosing)
**Zaleplon** Biomarkers: Zaleplon; 5-oxo-zaleplon (aldehyde oxidase product);	Plasma; urine	HPLC/LC–MS Detection window Plasma: hours Urine: <24h (typical metabolites)	~1–10 ng/mL (method dependent; short detection window)	Very short detection window; metabolites are inactive; parent drug detection indicates very recent dosing
**Zopiclone/Eszopiclone** **Biomarkers:** Zopiclone/eszopiclone; N-oxide; N-desmethyl zopiclone; 2-amino-5-chloropyridine	Plasma; urine; hair	LC–MS/MS/HRMS Detection window Plasma: hours Urine: ~1–3 days (metabolites)	~0.5–10 ng/mL (LOQ range in modern LC–MS/MS methods)	Moderate detection window; presence of N-oxide metabolite may be informative in forensic screens

**Table 11 metabolites-15-00745-t011:** Approved clinical uses of cannabinoids.

Product Name	Active Ingredient(s)	Regulatory Agency	Approved Indications
**Epidyolex^®^**	CBD	FDA EMA	Treatment of seizures associated with Lennox-Gastaut syndrome and Dravet syndrome in patients ≥ 2 years old. EMA also approves it for Tuberous Sclerosis Complex, in combination with other antiepileptics
**Dronabinol (Marinol^®^, Syndros^®^)**	Synthetic Δ^9^-THC	FDA	Appetite stimulation in patients with AIDS-related anorexia. Nausea and vomiting associated with chemotherapy in patients unresponsive to conventional antiemetics
**Nabilone (Cesamet^®^)**	Synthetic cannabinoid (THC analog)	FDA	Chemotherapy-induced nausea and vomiting in patients unresponsive to conventional antiemetics
**Nabiximols (Sativex^®^)**	THC + CBD (1:1, plant-derived)	EMA	Treatment of moderate to severe spasticity in multiple sclerosis, in patients who have not responded to other antispastic medications

**Table 12 metabolites-15-00745-t012:** Cannabinoids and their metabolites detected in blood and urine samples [249,250,251,252,253,254,255,256,257,258].

Cannabinoid Metabolite	Matrix (Blood, Urine)	Detection Method(s)	Typical LOD/LOQ (or Range)	Notes/Comments
**Δ^9^-THC**	Blood, Urine	GC–MS/MS with on-line SPE	LOD~0.15 ng/mL; LOQ~0.3 ng/mL	Validated for both blood and urine
**11-OH-THC**	Blood, Urine	GC–MS/MS with SPE	LOD~0.15 ng/mL; LOQ~0.3 ng/mL	Major active metabolite of THC
**THC-COOH**	Blood, Urine	GC–MS/MS with SPE	LOD~1.0 ng/mL; LOQ~3.0 ng/mL	Often measured as glucuronide conjugate in urine
**CBD**	Blood, Urine	GC–MS/MS with SPE	LOD~0.15 ng/mL; LOQ~0.3 ng/mL	Used to support interpretation of intake of CBD-rich products
**CBN**	Blood, Urine	GC–MS/MS with SPE	LOD~0.10 ng/mL; LOQ~0.2 ng/mL	Minor cannabinoid, sometimes marker of degradation or aging of product
**Δ^9^-THC, 11-OH-THC, THC-COOH, CBD, CBN, etc. (9 analytes)**	Urine	LC–MS/MS with SPE (minimal sample prep)	LOD~1 µg/L for non-carboxylated analytes and 5 µg/L for carboxylated analytes	Simultaneous detection without hydrolysis; rapid protocol
**Synthetic cannabinoid metabolites (up to 61 analytes)**	Urine	LC–MS/MS with SPE	LOD between 0.025 and 0.5 ng/mL	Validated panel for synthetic cannabinoids; high-throughput forensic tool
**Δ^9^-THC, 11-OH-THC, THC-COOH, CBD, CBN, etc. (broad panel)**	Plasma	HPLC-MS/MS LC–MS/MS (online extraction)	LOQ range~0.78–7.8 ng/mL	Method for 17 cannabinoids and metabolites
**THC, 11-OH-THC, THC-COOH, CBD, CBG, THCV, CBN**	Urine, Plasma	HPLC-MS/MS with hydrolysis	LOD < 1 ng/mL for many analytes	Method includes glucuronide hydrolysis for total analytes
**Δ^9^-THC, 11-OH-THC, THC-COOH, Δ^8^-THC, etc.**	Urine	LC–MS/MS (chromatographic separation of isomers)	LLOQ = 10 ng/mL for many analytes	Method distinguishes Δ^8^ vs. Δ^9^ isomers
**THC, 11-OH-THC, THC-COOH, THCAA, CBN, CBG, THCV, THC-glucuronide, THCCOOH-glucuronide**	Urine	LC–MS/MS with WAX-S pipette extraction	Linear ranges: 0.5–100 µg/L	Comprehensive urinary panel (11 analytes)
**THC, 11-OH-THC, THC-COOH, CBD, CBN, THC-Gluc, THCCOOH-Gluc, etc. (11-analyte panel)**	Urine	LC–MS/MS	LOQ~0.3–1.0 ng/mL (free and conjugated forms)	Differentiates dronabinol from cannabis use via metabolite profiles; detects minor cannabinoids
**THC metabolites (urine)**	Urine	LC–MS triple quadrupole (SLE extraction)	LOQ~5 ng/mL (for some THC metabolites)	According to Shimadzu application note
**THC, CBD, CBN, 11-OH-THC, THC-COOH (ultra-trace)**	Urine	UHPLC–MS/MS	LOD~0.002–0.008 ng/mL (free forms); ~0.005–0.017 ng/mL (total)	Very sensitive method for low exposure or secondhand smoke

Δ^9^-THC—Delta-9-tetrahydrocannabinol (main psychoactive cannabinoid); Δ^8^-THC—Delta-8-tetrahydrocannabinol; 11-OH-THC—11-Hydroxy-tetrahydrocannabinol; THC-COOH—11-nor-9-carboxy-tetrahydrocannabinol; CBD—Cannabidiol; CBN—Cannabinol; CBG—Cannabigerol; THCV—Tetrahydrocannabivarin; GC–MS/MS—Gas Chromatography coupled with Tandem Mass Spectrometry; LC–MS/MS—Liquid Chromatography coupled with Tandem Mass Spectrometry; LOD—Limit of Detection; LOQ—Limit of Quantification; SPE—Solid Phase Extraction; HPLC—High Performance Liquid Chromatography; WAX-S—Weak Anion Exchange Sorbent; SLE—Supported Liquid Extraction; SIM mode—Selected Ion Monitoring mode; UHPLC—Ultra-High Performance Liquid Chromatography.

**Table 13 metabolites-15-00745-t013:** Metabolic pathways and main metabolites of some natural and synthetic cannabinoids.

Cannabinoid	Pharmacokinetic Properties	Major Metabolic Pathways and Key Metabolites	Notes
**Δ^9^-THC** **Dronabinol**	Tmax: Inhalation ≈ 0–0.25 h (minutes); Oral ≈ 1–3 h T½: Multi-phasic; initial distribution 1–4 h; terminal elimination ~25–36 h (highly variable) Bioavailability: Inhalation 10–35%; Oral ≈ 6–20% (low, subject to first-pass) Vd: Large (highly lipophilic, extensive tissue distribution)	Major: 11-OH-THC (active) → 11-COOH-THC (inactive, major urinary metabolite) Enzymes: Primarily CYP2C9, CYP2C19 and CYP3A4 (hydroxylation and subsequent oxidation); conjugation to glucuronides	11-COOH-THC is the primary urinary marker used in forensic/toxicology screening; 11-OH-THC is pharmacologically active and important after oral dosing; long detection windows in chronic users due to lipophilic storage and slow release; metabolite/parent ratios and route of administration can be informative.
**Δ^8^-THC**	Tmax: Inhalation—minutes; Oral ≈ 1–3 h T½: Likely similar to Δ^9^-THC (multi-phasic; terminal elimination in the order of days), but data are limited Bioavailability: Expected similar to Δ^9^-THC (low oral bioavailability)	Major: 11-OH-Δ8-THC (reported), further oxidation to corresponding carboxy metabolite Enzymes: Similar CYP-mediated hydroxylation pathways as Δ^9^-THC (CYP2C9/2C19/3A4)	Pharmacology and metabolism are similar to Δ^9^-THC but clinical/pharmacokinetic data are less well characterized; forensic assays for Δ^9^-THC metabolites may cross-react.
**CBD**	Tmax: Oral ≈ 1–4 h (variable with formulation and fed state) T½: Reported ~18–32 h (single dose; can be longer with chronic dosing) Bioavailability: Oral ≈ 6–19% (formulation-dependent); extensive first-pass metabolism Vd: Large, highly tissue-distributed	Major: 7-OH-CBD, 7-COOH-CBD; conjugated metabolites (glucuronides) Enzymes: CYP2C19, CYP3A4 (oxidation) and UGT enzymes (glucuronidation)	CBD shows large interindividual variability; major metabolites (7-COOH) are abundant in plasma/urine; CBD can inhibit and be affected by CYP enzymes and UGTs (drug–drug interactions). Therapeutic formulations (e.g., Epidiolex) have well-characterized PK.
**Nabiximols** **(THC + CBD; oromucosal spray)**	Tmax: Oromucosal/oral absorption—typically 1–4 h (variable; partial buccal absorption produces faster onset than oral alone) T½: Parent compounds show multi-phasic kinetics; effective half-lives in the range of ~1–2 days for combined exposure (variable) Bioavailability: Higher than plain oral for oromucosal administration but variable between users	Major: 11-OH-THC (from THC), 7-OH-CBD (from CBD) and subsequent carboxy metabolites and glucuronides Enzymes: CYP2C9, CYP2C19, CYP3A4, CYP2D6 (minor), and UGTs for glucuronidation	Licensed pharmaceutical product; oromucosal route reduces but does not eliminate first-pass metabolism; predictable composition (standardized THC:CBD ratio) facilitates therapeutic monitoring.
**AB-PINACA**	Tmax: Smoking/inhalation—rapid (minutes); Oral Tmax may be 0.5–2 h depending on formulation T½: Reported short to moderate plasma half-lives (often a few hours for parent compound); terminal metabolites can persist longer Bioavailability: Variable; parent compound often rapidly metabolized	Major: Carboxylated and hydroxylated metabolites (e.g., AB-PINACA-COOH and multiple hydroxylated products) Enzymes/Pathways: Hydrolysis (CES1 reported) and oxidative metabolism (hydroxylation) on the alkyl/amino side chains; phase II conjugation	Parent drug is frequently absent or at low levels in urine; detection relies on specific metabolites (carboxy and hydroxylated derivatives). Novel synthetic cannabinoids show structural variability that complicates routine screening.
**AB-FUBINACA**	Tmax: Rapid after inhalation; Oral Tmax variable (often <2 h) T½: Parent compound short (hours); metabolites may be detectable for longer periods Bioavailability: Variable; rapid metabolism reduces parent exposure	Major: AB-FUBINACA-COOH and multiple hydroxylated metabolites; phase II conjugates Enzymes/Pathways: Hydrolysis by CES1 and oxidative metabolism on the indazole/alkyl side chains; conjugation for renal excretion	Forensic/toxicology screens target the carboxylated and hydroxylated metabolites rather than parent drug; potency and toxic effects can be considerably higher than phytocannabinoids.

**Table 14 metabolites-15-00745-t014:** Pharmacokinetic characteristics, analytical biomarkers, detection matrices, regulatory cut-off limits, and major analytical challenges of cocaine.

Substance	Pharmacokinetic Features	Biomarkers of Use/Recommended Matrices and Analytical Methods	Urine Cut-Off Limits (Screening/Confirmatory, ng/mL)	Major Analytical Challenges
**Cocaine**	The hydrochloride salt is effectively absorbed across mucous membranes (e.g., nasal or oral), whereas the freebase form (“crack”) is volatile and suitable for inhalation. Bioavailability: i.v.—100%; inhalation (smoked) ~70–90%; intranasal ~60–80% After intranasal administration, peak plasma concentrations (Cmax) are reached later (30–60 min) than after intravenous or smoking administration (1–5 min) Elimination half-life (t½): i.v. and smoked ~40–65 min; intranasal ~50–80 min Undergoes extensive metabolism in the liver and plasma: esterase-mediated hydrolysis to BE and EME; oxidative N-demethylation via CYP3A4, yielding norcocaine (hepatotoxic metabolite) Excreted renally as parent (1–5%) and inactive metabolites (BE, EME)	Biomarkers: cocaine (parent), BE, EME, anhydroecgonine methyl ester, cocaethylene Matrices: urine (detectable for several days), blood/plasma (reflects impairment, hours), oral fluid (recent exposure), hair (long-term monitoring) Methods: immunoassay screening followed by LC-MS/MS or GC-MS confirmation	Screen: 300 ng/mL (cocaine metabolites) Confirmatory: 100–150 ng/mL (cocaine metabolites)	Cross-reactivity in immunoassays (structurally similar compounds) Route specific metabolites Post-collection hydrolysis and derivatization steps to improve chromatographic resolution Specimen type influences interpretive conclusions

**Table 15 metabolites-15-00745-t015:** Principal cocaine-related analytes routinely targeted in confirmatory LC-MS/MS assays.

Primary Molecule	Metabolite(s) or Related Analytes	Diagnostic Relevance
**Cocaine**	Benzoylecgonine (BE)	Primary urinary biomarker; persists longest
	Ecgonine methyl ester (EME)	Marker of enzymatic hydrolysis by plasma cholinesterases
	Norcocaine	Active metabolite indicating hepatic oxidation
	Anhydroecgonine methyl ester (AEME)	Biomarker of crack cocaine smoking
	Cocaethylene	Marker of concurrent ethanol consumption
	Ecgonine	Secondary hydrolysis product, less specific

AEME = Anhydroecgonine methyl ester, a compound uniquely produced during pyrolysis of cocaine base (“crack”), considered a definitive marker of smoked cocaine.

**Table 16 metabolites-15-00745-t016:** Principal Ketamine-related analytes routinely targeted in confirmatory LC-MS/MS assays.

Analyte	Relevance
**Ketamine**	Parent compound; short detection window
**Norketamine (NK)**	Primary active metabolite; extends detection window
**Hydroxynorketamine (HNK)**	Antidepressant biomarker; low psychotomimetic activity
**Dehydronorketamine**	Minor metabolite; limited clinical significance
**Glucuronide conjugates**	Indicate metabolic clearance

**Table 17 metabolites-15-00745-t017:** Pharmacokinetic characteristics, analytical biomarkers, detection matrices, regulatory cut-off limits, and major analytical challenges of ketamine.

Substance	Pharmacokinetic Features [333,334,335]	Biomarkers of Use/Recommended Matrices and Analytical Methods	Urine Cut-Off Limits (Screening/Confirmatory, ng/mL)	Major Analytical Challenges
**Ketamine**	Bioavailability: i.v.—100%; i.m.—90–95%; intranasal ~40–45% Volume of distribution of 3–5 L/kg and moderate plasma protein binding (25–50%) Elimination half-life (t½) is approximately 2–3 h for ketamine and 4–6 h for its primary metabolite norketamine Hepatic metabolism is the primary route of biotransformation, mediated mainly by CYP2B6, CYP3A4, and CYP2C9, producing norketamine as the major active metabolite, which is subsequently converted to dehydronorketamine and hydroxynorketamine (HNK) isomers. Ketamine and its metabolites are excreted mainly by the kidneys as glucuronide conjugates.	Biomarkers: ketamine (parent), NK, HNK Matrices: urine (detectable for several days), blood/plasma (reflects impairment, hours), oral fluid (recent exposure), hair (long-term monitoring) Methods: immunoassay screening followed by LC-MS/MS or GC-MS confirmation	Screen: 50–100 ng/mL Confirmatory: 50 ng/mL	Cross-reactivity in immunoassays (structural related anesthetics) Enantiomer detection (S-ketamine) Cutting agents and impurities profiling

## Data Availability

The original contributions presented in this study are included in the article. Further inquiries can be directed to the corresponding author.

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
