# Peer review of "Differentiation of Therapeutic and Illicit Drug Use via Metabolite Profiling"

_metabolites, 2025, doi:10.3390/metabo15110745_

Round 1
Reviewer 1 Report
Comments and Suggestions for Authors
The manuscript presents a comprehensive review of strategies for distinguishing therapeutic from illicit drug use through metabolite profiling. It addresses an important and timely topic, given the global increase in controlled substance misuse and the challenges faced in forensic and clinical toxicology. The review is well-organized, scientifically sound, and provides valuable reference material for both clinical and forensic applications. However, there are several issues that have to be addressed:
- The manuscript is too extensive and highly informative, thus, the level of detail at times resembles a textbook chapter rather than a focused scientific review. Consider condensing repetitive background sections and emphasizing comparative interpretation of metabolite profiles across substance classes.
- The introduction could more explicitly state the research question or review objective in one concise sentence.
- The authors should more clearly delineate the novel analytical trends or emerging metabolomic approaches that advance the field beyond classical LC–MS/MS or GC–MS methods.
- While analytical principles are well described, a concise comparative table summarizing key biomarkers, matrices, detection limits, and analytical challenges would enhance readability and practical utility.
- The manuscript could better highlight how metabolic ratios, chiral analysis, and secondary markers translate into real-world interpretive decision-making — particularly in cases of therapeutic compliance versus illicit use.
- A few very recent sources (2024–2025) could be incorporated to strengthen the discussion on advanced omics techniques and machine learning in metabolite profiling.
- The authors should check for consistency in terminology (e.g., “therapeutic use” vs. “legitimate use”) and ensure uniform use of abbreviations after first mention.
- Several figure captions and chemical schemes (e.g., Figures 1–5) would benefit from simplification and improved visual clarity.A careful read-through for typographical consistency is advised.
Author Response
Dear Reviewer,
Thank you for your comments and suggestions. Here are our answers.
Comment 1:
The manuscript is too extensive and highly informative, thus, the level of detail at times resembles a textbook chapter rather than a focused scientific review. Consider condensing repetitive background sections and emphasizing comparative interpretation of metabolite profiles across substance classes.
Response 1:
Redundant background sections were removed, and greater emphasis was placed on the metabolic profiles of the different substance classes.
Comment 2:
The introduction could more explicitly state the research question or review objective in one concise sentence.
Response 2:
The review objective was added at the end of the Introduction.
Comment 3:
The authors should more clearly delineate the novel analytical trends or emerging metabolomic approaches that advance the field beyond classical LC–MS/MS or GC–MS methods.
Response 3:
The novel analytical trends or emerging metabolomic approaches that advance the field beyond classical LC–MS/MS or GC–MS methods have been discussed in the text.
Comment 4:
While analytical principles are well described, a concise comparative table summarizing key biomarkers, matrices, detection limits, and analytical challenges would enhance readability and practical utility.
Response 4:
Tables summarizing the key pharmacokinetic characteristics, analytical biomarkers, detection matrices, regulatory cut-off limits, and major analytical challenges of the selected opioids have been incorporated into the text.
Comment 5:
The manuscript could better highlight how metabolic ratios, chiral analysis, and secondary markers translate into real-world interpretive decision-making — particularly in cases of therapeutic compliance versus illicit use.
Response 5:
An additional was added to illustrate how metabolic ratios, enantiomeric composition, and secondary biomarkers are integrated into practical toxicological interpretation.
Comment 6:
A few very recent sources (2024–2025) could be incorporated to strengthen the discussion on advanced omics techniques and machine learning in metabolite profiling.
Response 6:
We have included recent information on the application of machine learning in metabolite profiling throughout the revised sections.
Comment 7:
The authors should check for consistency in terminology (e.g., “therapeutic use” vs. “legitimate use”) and ensure uniform use of abbreviations after first mention.
Response 7:
A check for consistency in terminology was made and corrections were made where necessary.
Comment 8:
Several figure captions and chemical schemes (e.g., Figures 1–5) would benefit from simplification and improved visual clarity. A careful read-through for typographical consistency is advised.
Response 8:
All figure captions have been simplified for improved clarity and readability. Minor visual and typographical adjustments were made to enhance consistency across all figures. Each figure now presents a more concise and standardized depiction of the respective metabolic or analytical pathway, in line with the reviewer’s recommendation.
All the changes in the manuscript are highlighted in yellow.
Reviewer 2 Report
Comments and Suggestions for Authors
This review by Stanila Stoeva-Grigorova et al. systematically reviews strategies and methods for distinguishing legal therapeutic use from illegal abuse of drugs through metabolite analysis, covering major drug categories such as opioids, amphetamines, benzodiazepines, cannabinoids, cocaine, and ketamine. Integrating parent drug-to-metabolite ratios, chiral analysis, detection of synthetic impurities, and multi-matrix sample analysis can significantly enhance the accuracy of conclusions. The topic of the paper holds significant clinical, forensic, and public health relevance. The content is detailed, and the references are extensive and up-to-date, reflecting the author's in-depth understanding of the field. Several revisions and improvements are needed in the following areas:
- The article primarily focuses on drug metabolites, but these metabolites are often eliminated quickly, making it difficult to detect them at times. In contrast, changes in endogenous substances may persist for longer periods. It is suggested to include more discussion on the application of metabolomics in distinguishing legal therapeutic use from illegal abuse.
- In Table 7, the authors summarized the pharmacokinetic properties of the main clinically used benzodiazepines and Z-drugs. It is suggested that the pharmacokinetic properties of opioids and amphetamine-type stimulants are also summarized and provided.
- Some sections of the article (e.g., opioids and cannabinoids) contain repetitive descriptions. It is recommended to further streamline the content to improve readability.
- It is suggested to include a summary table of metabolite ratio thresholds at the end of each chapter or in the conclusion section, allowing readers to quickly reference key judgment indicators.
- Further discussion on the metabolic characteristics and detection challenges of new psychoactive substances is recommended to enhance the timeliness and comprehensiveness of the review.
- It is advised to supplement the "Conclusion" section with a discussion on the limitations of metabolite analysis techniques, such as cost, technical barriers, and the complexity of data analysis.
Author Response
Dear Reviewer,
Thank you for your comments and suggestions. Here are our answers.
Comment 1:
The article primarily focuses on drug metabolites, but these metabolites are often eliminated quickly, making it difficult to detect them at times. In contrast, changes in endogenous substances may persist for longer periods. It is suggested to include more discussion on the application of metabolomics in distinguishing legal therapeutic use from illegal abuse.
Response 1: The role of metabolomics in differentiating legitimate therapeutic use from illicit abuse has been discussed in the text.
Comment 2:
In Table 7, the authors summarized the pharmacokinetic properties of the main clinically used benzodiazepines and Z-drugs. It is suggested that the pharmacokinetic properties of opioids and amphetamine-type stimulants are also summarized and provided.
Response 2:
Tables summarizing the key pharmacokinetic characteristics, analytical biomarkers, detection matrices, regulatory cut-off limits, and major analytical challenges of the selected opioids have been incorporated into the text.
Comment 3:
Some sections of the article (e.g., opioids and cannabinoids) contain repetitive descriptions. It is recommended to further streamline the content to improve readability.
Response 3:
The section on cannabinoids has been substantially revised and condensed to remove repetitive content and improve overall readability.
Comment 4:
It is suggested to include a summary table of metabolite ratio thresholds at the end of each chapter or in the conclusion section, allowing readers to quickly reference key judgment indicators.
Response 4:
Tables summarizing the key pharmacokinetic characteristics, analytical biomarkers, detection matrices, regulatory cut-off limits, and major analytical challenges of the selected opioids have been incorporated into the text.
Comment 5:
Further discussion on the metabolic characteristics and detection challenges of new psychoactive substances is recommended to enhance the timeliness and comprehensiveness of the review.
Response 5:
A discussion on the metabolic characteristics and detection challenges of new psychoactive substances was added.
Comment 6:
It is advised to supplement the "Conclusion" section with a discussion on the limitations of metabolite analysis techniques, such as cost, technical barriers, and the complexity of data analysis.
Response 6:
A discussion on the limitations of metabolite analysis techniques was added.
All the changes in the manuscript are highlighted in yellow.
